# Aerosol Vertical Mass Flux Measurements During Heavy Aerosol Pollution Episodes at a Rural Site and an Urban Site in the Beijing Area of the North China Plain

Renmin Yuan[1], Xiaoye Zhang[2, 4], Hao Liu[1], Yu Gui[1], Bohao Shao[1], Xiaoping Tao[5], Yaqiang Wang[2], Junting Zhong[2], Yubin Li[3] and Zhiqiu Gao[3]

[1]School of Earth and Space Sciences, University of Science and Technology of China, Anhui, 230026, China

[2]State Key Laboratory of Severe Weather & Key Laboratory of Atmospheric Chemistry of CMA, Chinese Academy of Meteorological Sciences, Beijing 100081, China

[3]School of Geography and Remote Sensing, Nanjing University of Information Science and Technology, Nanjing 210044, China

[4]Center for Excellence in Regional Atmospheric Environment, IUE, CAS, Xiamen 361021, China.

[5]School of Physical Sciences, University of Science and Technology of China, Anhui, 230026, China

*Correspondence*: Renmin Yuan (rmyuan@ustc.edu.cn) and Xiaoye Zhang (xiaoye@cma.gov.cn)

## Abstract:

Due to excessive anthropogenic emissions, heavy aerosol pollution episodes (HPEs) often occur during winter in the Beijing-Tianjin-Hebei (BTH) area of the North China Plain. Extensive observational studies have been carried out to understand the causes of HPEs; however, few measurements of vertical aerosol fluxes exist, despite them being the key to understanding vertical aerosol mixing, specifically during weak turbulence stages in HPEs. In the winter of 2016 and the spring of 2017 aerosol vertical mass fluxes were measured by combining large aperture scintillometer (LAS) observations, surface $PM_{2.5}$ and $PM_{10}$ mass concentrations, and meteorological observations, including temperature, relative humidity (RH), and visibility, at a rural site in Gucheng (GC), Hebei Province, and an urban site at the Chinese Academy of Meteorological Sciences (CAMS) in Beijing located 100 km to the northeast. These are based on the light propagation theory and surface-layer similarity theory. The near-ground aerosol mass flux was generally lower in winter than in spring and weaker in rural GC than in urban Beijing. This finding provides direct observational evidence for a weakened turbulence intensity and low vertical aerosol fluxes in winter and polluted areas such as GC. The HPEs included a transport stage (TS), an accumulative stage (AS), and a removal stage (RS). During the HPEs from 25 January 2017 to January 31, 2017, in Beijing, the mean mass flux decreased by 51% from 0.0049 mg m$^{-2}$s$^{-1}$ in RSs to 0.0024 mg m$^{-2}$s$^{-1}$ in the TSs. During the ASs, the mean mass flux decreased further to 0.00087 mg m$^{-2}$s$^{-1}$, accounting for approximately 1/3 of the flux in the TSs. A similar reduction from the TSs to ASs was observed in the HPE from 16 December 2016 to 22 December 2016 in GC. It can be seen that from the TS to the AS, the aerosol vertical turbulent flux decreased, but the aerosol particle concentration within surface layer increased, and it is inferred that in addition to the contribution of regional transport from upwind areas during the TS, suppression of vertical turbulence mixing confining aerosols to a

shallow boundary layer increased accumulation.

# 1 Introduction

Recently, due to the country's rapid development of industrialization and urbanization, China
has experienced heavy aerosol pollution episodes, particularly in the Beijing, Tianjin and Hebei
(BTH) region, which is one of the most polluted areas in China (Zhang et al., 2012). The pollution
episodes often last for a long duration in the BTH region and cover a wide area, particularly in
winter; they also severely reduce near-ground visibility (Lei and Wuebbles, 2013) and can have
detrimental effects on public health (He et al., 2018;Cao et al., 2012). This heavy pollution
environment has received extensive attention in recent years, and many observational studies have
been carried out (Zhong et al., 2018b; Sun et al., 2014; Wang et al., 2015; Guo et al., 2011; Zhang
et al., 2009b; Huang et al., 2014). Modelling studies have also been performed to examine the
regional transport of pollutants (Wang et al., 2014) and to study the important role of large-eddy
convective turbulent mixing in the vertical transfer of pollutants from a field campaign in Beijing
(Li et al., 2018). However, few study on the turbulence contribution of the aerosol turbulent flux in
the surface layer has been conducted.
Ground pollutant emissions are known as the main source of aerosols in the atmosphere.
However, in previous studies, no measurements of ground emissions during heavy pollution events
were collected. Surface emission data are currently required for model verification and pollution
predictions, and these data are primarily obtained through emission inventories (Wu et al., 2012;
Bond et al., 2004). The establishment of emission inventories is primarily based on emission activity
and emission factor (EF) data (Akagi et al., 2011; Lu et al., 2011; Roden et al., 2006; Zhang and
Tao, 2009). Emissions data are mainly obtained from statistical yearbooks (Zhang et al., 2009a).
Some studies have used fixed EFs while others have implemented dynamic EFs (Bond et al., 2004;
Zhang et al., 2009a). Many factors are considered in dynamic EFs, such as the size of a city, the
degree of economic development, the type of fuel, the kind of technology, product energy
consumption, the control technology, and so on, as well as estimates based on actual measured
meteorological parameters and aerosol parameters (Chen et al., 2015;Karvosenoja et al., 2008;Shen
et al., 2013). A numerical model has also been used to estimate average fleet emission factors in
typical urban conditions (Ketzel et al., 2003; Krecl et al., 2018). The uncertainties in the emissions
of primary aerosols for inventories are high due to the highly uncertain contributions from the
residential sector (Li et al., 2017), and the error in aerosol fluxes based on the use of emission
inventories is huge (Liu et al., 2017;Zheng et al., 2017). Emission inventories constructed using the
EF method provide only the total emission amount of atmospheric pollutants within a region.
However, the emission data should be gridded to a suitable scale for air quality modeling and
pollution predictions. Thus, near-surface aerosol emission data with a higher temporal and spatial
resolution are urgently needed.
Many methods have been used to obtain aerosol flux data. For the upward transport of aerosols
near the surface layer, the aerodynamic approach was adopted in the early years. The aerosol
concentration gradient at different heights was measured and then calculated based on the similarity
theory of the near-surface layer or calculated by the boundary layer box model, which can be based
on meteorological data (Ceburnis et al., 2016;Hourdin et al., 2015;Zhang and Li, 2014). The
emission rates of bioaerosols were also estimated from spore counts and molecular tracers (Elbert
et al., 2007). The abundance of microbes and meteorological data were measured, and an estimate
may be derived from the sea-air exchange of microorganisms (Mayol et al., 2014).

With the use of instruments for measuring the number of aerosol particles (for example, a
condensational particle counter, abbreviated as CPC by TSI), the eddy covariance (EC) method has
been applied, and measurements of the aerosol particle number flux have become possible (Buzorius
et al., 1998). The vertical turbulent flux of the aerosol particle number density $F_p$ is denoted as a
cross-covariance between the aerosol particle number concentration $N'$ and the vertical wind speed
$w'$ (Ripamonti et al., 2013). To obtain vertical turbulent flux of the aerosol number density, the EC
principle allows quantifying the number flux from fluctuation measurements. As a result, the
vertical turbulent flux of the aerosol particle number density has been measured in many cities, such
as in Toronto, Canada (Gordon et al., 2011), Stockholm, Sweden (Vogt et al., 2011b), Helsinki,
Finland (Ripamonti et al., 2013), London, UK (Harrison et al., 2012), the Blodgett Forest
Observatory in the United States (Farmer et al., 2011), and measurements of sea salt aerosol fluxes
in northern Europe (Brooks et al., 2009;Sproson et al., 2013). These results have shown the
quantitative relationship among urban aerosol fluxes, urban vehicle emissions, and meteorological
conditions (Jarvi et al., 2009) and have been used to determine transport characteristics of sea salt
aerosol and provide further knowledge of aerosol properties (Nemitz et al., 2009). These
measurements have been mainly collected in cities because of their anthropogenic contributions to
aerosol emissions. These data can be used as routine model inputs. Direct eddy covariance
measurements of aerosol exchanges in tropical forests, where primary biological aerosol particles
represent a substantial fraction of the airborne particulate matter (Graham et al., 2003), were also
performed by Ahlm et al. (Ahlm et al., 2010a;Ahlm et al., 2010b) and Whitehead et al. (Whitehead
et al., 2010), potentially giving a proxy for microbial emissions in tropical ecosystems.

Although measurements of urban aerosol particle number density fluxes have been collected,
the current eddy covariance method only provides fluxes for the aerosol particle number density at
a point. We know that the underlying surface of a city is very complex, and thus the aerosol particle
flux is not homogeneous in the horizontal. For a complex underlying surface such as a city, these
point measurements are not representative of wider area. Therefore, it is of great importance to
design an aerosol flux measurement system with larger spatial representation.

The use of eddy covariance principles to measure sensible heat fluxes has been widely
performed (Lee, 2004). Current sensible heat fluxes can also be obtained using a large aperture
scintillometer (LAS) based on the light propagation theory and atmospheric surface layer similarity
theory (Zeweldi et al., 2010). This configuration makes it possible to achieve aerosol mass flux
measurements using the same principles. Recently, we measured the imaginary part of the
atmospheric equivalent refractive index structure parameter based on the light propagation theory
(Yuan et al., 2015). The results showed that the imaginary part of the atmospheric equivalent
refractive index structure parameter is related to turbulent transport and the spatial distribution
characteristics of aerosols. Experiments also showed that there is a strong correlation between the
imaginary part of the atmospheric equivalent refractive index and the mass concentration of aerosol
particles (Yuan et al., 2016). Thus, similar to the temperature structure parameter reflecting the
sensible heat flux, the structural parameter of the imaginary part of the atmospheric equivalent
refractive index can reveal the mass flux of aerosol particles. This paper attempts to measure the
aerosol mass flux in the BTH area, especially during heavy aerosol pollution episodes.

Generally, based on the PM$_{2.5}$ daily mean mass concentration limit in the primary standard of
China's national environmental quality standards (EPD, 2012), a pollution episode is referred to as
the period during which the PM$_{2.5}$ concentration exceeds 80 μg m$^{-3}$ for 3 consecutive days between
two clean periods, while a period when the PM$_{2.5}$ level is less than 35 μg m$^{-3}$ is defined as a clean
period. Pollution episodes with peak PM$_{2.5}$ values of more than 400 μg m$^{-3}$ or less than 300 μg m$^{-3}$
are termed heavy-pollution episodes (HPEs) or light-pollution episodes (LPEs), respectively (Zhong
et al., 2017b).
To gain a deeper understanding of the interaction between atmospheric heavy pollution and
weather in the BTH region, joint observations have been carried out in the BTH region since the
winter of 2016 (Zhong et al., 2018c;Zhong et al., 2018b;Wang et al., 2018;Shen et al., 2018). Based
on meteorological causes of the increase or decrease in PM2.5 mass concentrations, an HPE in the
BTH region can be divided into a transport stage (TS), an accumulative stage (AS) and a removal
stage (RS). During the TS, the PM$_{2.5}$ high concentration levels are caused by relatively strong
southerly winds, which carry polluted air masses from more populated southern industrial regions
(Guo et al., 2014; Zhong et al., 2018a). Before elevation of PM$_{2.5}$ concentrations during TSs, the
urban PM$_{2.5}$ mass concentration of Baoding, which is typically representative of pollution conditions
in the south of Beijing, was much higher than Beijing; the winds in Beijing rapidly shifted from
northerly to southerly. Then the rising in PM$_{2.5}$ occurred, consistently with southerly slight or gentle
breezes in the BL. The southerly air mass moves more than 288 km d$^{-1}$ below 500 m (estimated
from the measured wind speed), which are fast enough to transport pollutants to Beijing. Such
processes indicate southerly pollutant transport is primarily responsible for elevation of PM$_{2.5}$
concentrations, given the pollution transport pathway of the southwest wind belt determined by the
unique geographic features of the North China Plain, with the Tai-hang Mountains and the Yan
Mountains strengthening the southwest wind belt and leading the convergence of pollutant transport
in Beijing (Su et al., 2004). During the ASs, PM$_{2.5}$ increase is dominated by stable atmospheric
stratification characteristic of southerly slight or calm winds, near-ground anomalous inversion, and
moisture accumulation. When aerosol particles by vertical transport are accumulated to a certain
degree, the dominant scattering aerosols will substantially back-scatter solar radiation, causing a
reduction in the amount of solar radiation that reaches the surface, which creates a near-ground
cooling effect resulting in the lack of vertical mixing in the near-surface layer. (Zhong et al., 2018c).
A feedback effect of further worsened meteorological conditions aggravates PM$_{2.5}$ pollution (Zhong
et al., 2017$a$). During the RSs, strong north-westerly winds whose velocity increases with height
occur predominantly. Strong northerly winds are from less populated north mountainous areas and
carry unpolluted air masses to Beijing, which is favorable for pollution dispersion. The observations
reveal the large-scale and mesoscale transport processes of aerosols between HPEs in the BTH
region in the winter of 2016. However, during HPEs, no research has been conducted in the BTH
area on quantifying the contribution of surface emissions to the concentration of pollutants. In this
study, we focus on aerosol emission during HPEs through field observations of aerosol turbulent
based on the light propagation theory and surface similarity in the Beijing urban district and
Gucheng suburban area.
The second section of this paper introduces the theory of aerosol vertical turbulent flux
measurements, the third section introduces the experiment, the fourth section gives the experimental
results, and finally, the conclusion and discussion are presented in the fifth section.

## 2 Theory and methods

The principles for calculating the vertical flux of aerosol particles and the approach for calculating the friction velocity and characteristic temperature using the temperature and wind profiles is presented in the following subsections.

**2.1 Calculation of the aerosol mass vertical flux**

According to the micrometeorological principle (Stull, 1988), similar to the estimation method of the sensible heat flux, the aerosol flux $F_a$ can be obtained as follows:

$$F_a = u_* M_* \tag{1}$$

where $u_*$ is the friction velocity, which can be obtained from the temperature and wind speed profiles or directly from three-dimensional wind speed measurements; see Sec. 2.2. Prior experiments have shown that the spectral characteristics of aerosol number concentration fluctuations approximate the spectral characteristics of molecular density fluctuations. (Martensson et al. 2006; Vogt et al. 2011b). Therefore, aerosol particles can be approximated as scalars for turbulent statistics, and characteristic parameters $M_*$ similar to the scalars can be introduced, which can be regarded as the atmospheric aerosol mass concentration scale in the surface layer and deduced from surface layer similarity theory. This approximation is similar to the surface-layer temperature scale (Stull, 1988) as follows:

$$\frac{C_M^2 (z-d)^{2/3}}{M_*^2} = \eta(\xi) \tag{2}$$

where $z$ is the measurement height, $d$ is the zero-displacement height (Evans and De Bruin, 2011; Hartogensis et al., 2003), $\xi=(z-d)/L$ is the nondimensional stability parameter, $L$ is the Monin-Obukhov (MO) length and defined as $L = \dfrac{\bar{T} u_*^2}{\kappa g T_*}$ (Stull, 1988), $\bar{T}$ is the average temperature, $T_*$ is the surface-layer characteristic temperature, $\kappa$ is the von Karman constant, which is 0.4, and g is acceleration due to gravity. The stability function ($\eta(\xi)$) can be expressed as follows depending on the stability condition (DeBruin et al., 1995):

$$\eta(\xi) = a_1 [1 - a_2 \xi]^{-2/3} \tag{3}$$

for unstable conditions ($\xi<0$), and the following:

$$\eta(\xi) = b_1[1 + b_2(\xi)^{e_1}] \qquad (4)$$


for stable conditions ($\xi >= 0$) (Wyngaard et al., 1971).
In Eqs. (3) and (4), $a_1$, $a_2$, $b_1$, $b_2$ and $e_1$ are constants, and different experiments have provided
different values, although the differences between these results are small. It is assumed that the
aerosol mass concentration fluctuation characteristics are the same as the temperature fluctuation
characteristics and the same similarity law of Eq. (2) is satisfied. Therefore, based on the
experimental data, the values of $\sqrt{\dfrac{C_T^2(z-d)^{2/3}}{\eta(\zeta)}}$ and $T_*$ are calculated using various schemes.
After comparing the differences between the two, the scheme of DeBruin et al (DeBruin et al., 1995)
with $a_1=4.9$, $a_2=9$, $b_1=5$, and $b_2=0$ is taken based on minimal difference using the experimental data
of this study.
$C_M^2$ in Eq. (2) is the aerosol mass concentration structure parameter. We assume that the
aerosol particles in the atmosphere follow the movement of the air and satisfy the turbulent motion
law. Previous studies have shown that the particle concentration fluctuation spectra follow a '-5/3'
power law under unstable stratification conditions (Martensson et al., 2006; Vogt et al., 2011b), and
the velocity-concentration co-spectra follows a '-4/3' power law (Martensson et al., 2006; Vogt et
al., 2011a; Kaimal et al., 1972). Thus, similarity of atmospheric aerosols and temperature can be
assumed for the purpose. Then, at a separation ($r$) of the order in the inertial subrange in a locally
isotropic field, the aerosol mass concentration (denoted as $M_a$) structure function ($D_M(r)$) follows a
"2/3 law" (Wyngaard, 2010) and can be expressed as $D_M(r) = \overline{[M_a(x) - M_a(x + r)]^2} = C_M^2 r^{2/3}$,
where $x$ is the position vector, $r$ is the separation vector, and the overbar indicates the spatial average.
The following describes the method to deduce the aerosol mass concentration structure
parameter $C_M^2$.
Although the aerosol particles are dispersed in the air, the macroscopic behavior of the gas-
particle two-phase mixture is the same as if it is perfectly continuous in structure and physical
quantities, such as the mass and refractive index associated with the matter contained within a given
small volume, which will be regarded as being spread continuously over that volume. The aerosol
particles and gases in the atmosphere can be considered as an equivalent medium, and an
atmospheric equivalent refractive index (AERI) $n_{equ}$ is introduced that contains the real part $n_{re}$ and
the imaginary part $n_{im}$ of the equivalent refractive index. Thus, $n_{equ}=n_{re}+i \bullet n_{im}$. For visible light, the
attenuation of light by gases in the atmosphere is very weak; the cause of the attenuation is the
absorption and scattering due to aerosol particles. Therefore, the real part of the equivalent medium
of aerosol particles and gases is determined by the gas composition of the air. The fluctuation of the
real part is mainly determined by temperature fluctuations; the imaginary part is determined by the
aerosol particles, and the fluctuation of the imaginary part is determined by fluctuations in the
aerosol concentration.

For visible light, there is a robust linear relationship between the variation of the real part of

the AERI and the variation of the atmospheric temperature, namely, $R_{TN} = \dfrac{\delta T}{\delta n_{Re}}$ ; thus, we have the
following:

$$R_{TN} = -1.29 \times 10^{4} \times (1 + \frac{7.52 \times 10^{-3}}{\lambda^{2}})^{-1} \frac{\overline{T}^{2}}{\overline{P}} \tag{5}$$

which is based on the relationship between the real part of the AERI ($n_{Re}$) and atmospheric
temperature (Tatarskii, 1961). Because the wavelength is deterministic, the ratio $R_{TN}$ can be obtained
by measuring the atmospheric temperature. The imaginary part of the AERI ($n_{Im}$) has a close
correspondence with the extinction coefficient of the equivalent medium, and the extinction
coefficient is inversely proportional to the visibility. The light wavelength is selected as 0.620 μm.
This wavelength is only weakly absorbed by $O_3$; therefore, the observed absorption is primarily due
to aerosol (Brion et al., 1998; Lou et al., 2014; Nebuloni, 2005). Higher concentrations of aerosols
in the atmosphere are related to lower visibility and vice versa; thus, the relationship between the
imaginary part of the AERI and the atmospheric aerosol mass concentration can be established. The
ratio of the atmospheric aerosol mass concentration to the imaginary part of the AERI $R_{MN}$ can be
defined as follows:

$$R_{MN} = \frac{M_a}{n_{Im}} . \tag{6}$$

Theoretical analysis has revealed that $R_{MN}$ is associated with the aerosol particle size

distribution, mass density of the aerosol particles, and the aerosol particle refractive index. Because
of the relatively small variations in particle size distributions and aerosol refractive index (Dubovik
et al., 2002), $R_{MN}$ can be treated as a constant for surface-layer aerosols at a given location. Of course,
$R_{MN}$ can be obtained by simultaneously measuring $M_a$ and the imaginary part of the AERI, so that
real-time $R_{MN}$ can be obtained. $M_a$ approximates the PM$_{10}$ value. The variable $n_{im}$ can be calculated
as follows (Yuan et al., 2016):

$$n_{Im} = \frac{0.55e-6}{4\pi} \cdot \frac{3.912}{L_V} \tag{7}$$

where the unit of visibility ($L_V$) is m.
According to Eqs. (5) and (6), we have the following:
$$C_T^2 = R_{TN}^2 C_{n.\mathrm{Re}}^2 \tag{8}$$

$$C_M^2 = R_{MN}^2 C_{n,\mathrm{Im}}^2 \tag{9}$$

Thus, the temperature structure parameter $C_T^2$ and the aerosol mass concentration fluctuation
structure parameter $C_M^2$ are converted into the measurement of the real and imaginary structural
parameters of the AERI, namely, $C_{n,\mathrm{Re}}^2$ and $C_{n,\mathrm{Im}}^2$ respectively.
The measurement of relevant parameters is performed based on the light propagation theory.
When light is transmitted in an equivalent medium, the AERI fluctuation will cause light
fluctuations in light intensity. When the attenuation caused by scattering and absorption along the
propagation path is very weak, light intensity fluctuation depends on the fluctuation of the real part
of the AERI along the propagation path. When the attenuation caused by scattering and absorption
along the propagation path is relatively strong, the light intensity fluctuation is also related to the
fluctuation of the imaginary part of the AERI along the propagation path. With the spectral analysis
method, the LAS light intensity fluctuations can be separated into the contributions of the real and
imaginary parts of the AERI. The contribution of the real part of the AERI corresponds to the high
frequencies, whereas the contribution of the imaginary part of the AERI corresponds to the low
frequencies, suggesting that the variances resulting from the real and imaginary parts are
independent. Therefore, the light intensity variances induced by the real and imaginary parts can be
detected separately at high frequencies and low frequencies from the LAS measurements (Yuan et
al., 2015). Thus, the real and imaginary structure parameters of the AERI can be calculated by our
developed LAS.
So far, we have completed the estimation of the aerosol mass turbulent flux.
According to the previous derivation and analysis, there are two calculation schemes for
determining the aerosol mass flux as follows:
$$F_{a1} = (\frac{C_{n,\mathrm{Im}}^2}{C_{n.\mathrm{Re}}^2})^{1/2} \frac{R_{MN}}{R_{TN}} u_* |T_*| \tag{10}$$

$$F_{a2} = u_* \sqrt{\frac{C_M^2 (z-d)^{2/3}}{\eta(\xi)}} = u_* R_{MN} \sqrt{\frac{C_{n,\mathrm{Im}}^2 (z-d)^{2/3}}{\eta(\xi)}} \tag{11}$$

When the free convection approximation ($-\xi \gg 1$) is assumed, based on the definition of the

M-O length, and the similarity theory (Wyngaard et al., 1971), the following can be obtained:


$$F_{a3} = a(\frac{g}{\overline{T}})^{1/2} R_{TN}^{1/2} (C_{n,\mathrm{Re}}^2)^{1/4} R_{MN} (C_{n,\mathrm{Im}}^2)^{1/2} (z-d) \qquad (12)$$

where the coefficient $a = a_1^{-3/4} a_2^{1/2} \kappa^{1/2}$ can be taken as 0.567 (DeBruin et al., 1995; Lagouarde

et al., 2006). Eqs. (10)-(12) are the theoretical basis for the aerosol mass flux measurements.

According to Eqs. (10)-(12), the vertical turbulent flux of aerosol particles is related to the

strength of turbulent fluctuations of temperature and aerosol mass concentration fluctuations.

In this study, there is a weather tower in the north of Beijing. The weather tower is 6.1km far

from the CAMS site. The meteorological observation data from the weather tower show that

applicability of the Monin-Oubhov similarity theory under stable condition causes a significant error

for $T_*$ or $u_*$, while the Monin-Oubhov similarity theory is still basically applicable in the case of

unstable stratification (Liu et al. 2009). In the roughness sub-layers of other cities, under the

condition of unstable stratification, the local similarity theory is similar to MOST (Zou et al. 2018,

2019). Because the height of the LAS instrument at the CAMS site was 43 m, during most of the

time the conditions assumed for free convection were easily satisfied. During the day, the surface

layer is usually unstable. At night, for the city, even if there is an inversion at a higher altitude, due

to the existence of the urban heat island, the surface layer is often weakly unstable. The stable

stratification situation is rare (Li et al., 2007). Therefore, aerosol fluxes in Beijing are calculated

using Eq. (12) based on the assumption of free convection.

Based on the discussion above, the LAS technique is capable to determine the magnitude of

the flux but not the sign. In general, the aerosols are very heterogeneous in space and the measured

fluxes show typically large variation in magnitude including the sign. Over the polluted areas, which

behave as the source, the emissions presumable overwhelmingly exceed the deposition sinks

(Ripamonti et al., 2013). Therefore, a rough quantification of the deposition sink would allow

concluding that the sink term is negligible and the flux quantified by LAS can be assumed to

represent the upward fluxes.

**2.2 Calculation of the friction velocity and surface-layer characteristic temperature**

To calculate the aerosol vertical turbulent flux, according to Eq. (10), the values of the friction

velocity $u_*$ and the characteristic temperature $T_*$ are required. These can be obtained via wind speed

and temperature profile data. From the near-surface similarity theory, the temperature and wind

speed data measured at two heights of $z_1$ and $z_2$ can be used in the expressions of the friction velocity

$u_*$ and the characteristic temperature $T_*$ (Stull, 1988) as follows:

$$u_* = \frac{\kappa[U(z_2) - U(z_1)]}{\ln\dfrac{z_2}{z_1} - \Psi_U(\xi_2) + \Psi_U(\xi_1)} \qquad (13)$$

$$T_* = \frac{\kappa[T(z_2) - T(z_1)]}{0.74[\ln\dfrac{z_2}{z_1} - \Psi_T(\xi_2) + \Psi_T(\xi_1)]} \qquad (14)$$

where U($z_1$) and U($z_2$) are the measured velocities at heights $z_1$ and $z_2$, respectively, T($z_1$) and T($z_2$) are the measured temperatures at heights $z_1$ and $z_2$, respectively, $\xi_1$ and $\xi_2$ are the stabilities at heights $z_1$ and $z_2$, respectively, and $\Psi_U$ and $\Psi_T$ are the correction terms for the velocity and temperature profiles under the condition of stability $L$. Under unstable conditions (Stull, 1988), we have the following:

$$\Psi_U(\xi) = \ln[(\frac{1+x^2}{2})(\frac{1+x}{2})^2] - 2\arctan(x) + \frac{\pi}{2}, \quad x = (1-15\xi)^{1/4} \tag{15}$$

$$\Psi_T(\xi) = \ln[(\frac{1+y}{2})^2], \quad y = (1-9\xi)^{1/2} \tag{16}$$

Under stable conditions (Cheng and Brutsaert, 2005), we have the following:

$$\Psi_U(\xi) = -a\ln[\xi + (1+\xi^b)^{1/b}], \quad a\text{=6.1, } b\text{=2.5.} \tag{17}$$

$$\Psi_T(\xi) = -c\ln[\xi + (1+\xi^d)^{1/d}], \quad c\text{=5.3,} d\text{=1.1.} \tag{18}$$

Based on Eqs. (13)-(18), the friction velocity $u_*$ and characteristic temperature $T_*$ can be determined.

## 3 Measurements and data processing

### 3.1 Introduction of Experiments

Observations were collected at two locations (two rectangles in Fig. 1a) from December 2016 to March 2017: a rural site in Gucheng (GC site), Hebei Province and an urban site at the Chinese Academy of Meteorological Sciences (CAMS site) in Beijing. The distance between the two locations is approximately 100 km. According to the theoretical methods defined in the preceding section, to estimate the aerosol turbulent flux, the ratio of the aerosol mass to the imaginary part of the AERI, the ratio of the temperature to the real part of the AERI, the real and imaginary parts of the atmospheric equivalent refractive index structure parameter (AERISP, $C_{n,Re}^2$ and $C_{n,Im}^2$), the friction speed, and the characteristic temperature must all be obtained. If the free convection condition is satisfied, fewer parameters are required, including the real and imaginary parts of the AERISP, the ratio of the aerosol mass to the imaginary part of the AERI, the ratio of the temperature to the real part of the AERI, and the atmospheric temperature.

Two sets of LASs developed by our research group were installed at the top of the building of the Chinese Academy of Meteorological Sciences (point A in Fig. 1b) and at the top of a two-story building in the farm of the Central Meteorological Bureau of Gucheng Town, Baoding City (point D in Fig. 1c). The light intensity sampling frequency of the receiving end was 500 Hz, and a file was recorded every 20 minutes. Then, the real and imaginary parts of the AERISP were calculated. In the CAMS site, the transmitter end of the LAS was placed on the roof of a building on the east side of the Chinese Academy of Meteorological Sciences, and the receiver end was placed at the top of the Chinese Academy of Meteorological Sciences. The propagation path was along an east-west direction. The distance between the two ends was 550 meters as shown in Fig. 1d. The light beam passed over urban buildings, residential areas and urban roads. The beam height was 43 meters. The average height of the building below the beam was 24 meters; thus, the zero-displacement was 18 meters (24 * 0.67 = 18) (Leclerc and Foken, 2014), and the effective height of the beam was 25

meters. At the Beijing observation point, the conventional meteorological parameters are measured
on the same roof, 20 meters away from the receiving end and in the northwest direction of the
receiving end. The measurement heights were 1.5 m and 10 m above the roof for air temperature
and wind speed. To calculate the aerosol flux, it is necessary to obtain the ratio of the aerosol mass
to the imaginary part of AERI and to measure the aerosol mass concentration and visibility. In
Haidian District, there is a site to measure the visibility of the near-surface layer (point B in Fig.
1b), and the $PM_{10}$ mass concentration measurements were collected at Guanyuan Station (see point
C in Fig. 1B). The sampling interval for the visibility and $PM_{10}$ mass concentration measurements
was 1 h. The measurement height of points B and C in Fig. 1b was approximately 20 metres. The
ratio of the aerosol mass $PM_{10}$ to the imaginary part of the AERI was calculated based on the data.
The measurements were collected at the CAMS site from 15 January 2017 to 20 March 2017.
In the GC site (point D in Fig. 1c, namely, the LAS position) of Gucheng, Baoding, Hebei, the
transmitter of the LAS was placed on the roof of a two-story building with a height of 8 m, and the
receiving end was located in a room in a three-story building on the west side of National Highway
107 at the same height as the transmitting end. The distance between the transmitting end and the
receiving end was 1670 metres. The terrain between the transmitting end and the receiving end was
flat, with farmland, a national road and sporadic trees below the beam, as seen in Fig. 1e. Near the
light beam, there was a 30-meter-high meteorological observation tower, in which the temperature,
relative humidity (RH), and wind speed were measured at five levels (1 m, 3 m, 8 m, 18 m, and 28
m). The friction speed and characteristic temperature were calculated according to the temperature
wind speed profile. Visibility observations were made in Xushui District near the LAS position (see
point E in Fig. 1c). The $PM_{10}$ mass concentration was measured in Beishi District (see point F in
Fig. 1c). From Fig. 1c, the three observation points (points D, E and F in Fig. 1c) formed a nearly
straight line and were distributed in a northeast-southwest direction. During the experimental
observation period, a northeast-southwest wind prevailed; thus, the Xushui District visibility data
and Beishi District $PM_{10}$ data can approximate the situation of the scintillometer position. The
measurements were collected at the GC site from 17 November 2016 to 30 March 2017.
**3.2 Data quality control**
There are two types of variables, namely mean variables and fluctuation variables. Mean
variables include temperature, wind speed, wind direction, $PM_{10}$, and visibility for averages of 30
minutes or 60 minutes. Data quality control for the mean variables was conducted by comparing the
measured data at different heights or at different stations. Same variables between different heights
and different locations having the same trend are considered high quality. All the measured mean
data were determined to be adequate. Fluctuation variables include the high-frequency intensity
fluctuation data measured by the LAS, the real and imaginary parts of the AERISP, and the
calculated aerosol flux. Quality control mainly consists of the elimination of spike and
supplementing missing data.
Peaks in the light intensity fluctuation data appear because the received signal quickly increases
when the light signal is blocked, such as due to birds along the transmission path. The data
processing program automatically determines this situation. When this happens, the current 20-
minute period is rejected. For the real and imaginary parts of the AERISP and the aerosol flux data,
(a) 3 times the standard deviation (SD) of the anomaly and (b) 3 times the SD of the difference

between adjacent moments (AMD) were determined. A trend of two-hour averages, namely, 6-point moving averages, is first obtained. Then, the difference between the measured value and the trend at each moment was calculated, and the mean and SD of the difference were also calculated. The data with differences from the trend exceeding 3 times the SD were considered as spikes. The method for judging 3 times the SD of the AMD was first to calculate the AMD and then calculate the mean and SD of the AMDs. Any data whose AMD deviated from the mean of the AMD by more than 3 times the SD of the AMD was considered an error. Less than 5% of the data were found to contain spikes or errors.

The data determined to be errors were supplemented with the average of the nearby observations. Of course, if data were missing over a long period, the missing gap could not be filled. For this situation, further gap-filling was not considered.

Other errors exist in the measurements using a LAS due to specific reasons (Moene et al., 2009); for example, the impact of the deviation of the shape of spectrum from von Karman's scheme and the intermittent variations in the characteristics of that spectrum on the LAS signal were not considered in this study.

# 4 Results

First, the visibility and $PM_{10}$ aerosol mass concentration results at the CAMS site and the GC site are given and compared. Then, the characteristics of aerosol transport in typical weather conditions at the CAMS site and the GC site are discussed. Finally, the aerosol flux characteristics during the HPEs are analyzed.

## 4.1 Relationship between $n_{im}$ and $PM_{10}$

To obtain the ratio of the atmospheric aerosol mass concentration to the imaginary part of the AERI ($n_{im}$) $R_{MN}$, $PM_{10}$ and visibility were measured.

The maximum $PM_{10}$ concentration in the Baoding area appeared at 1:00 on January 28, 2017 (up to 1071 μg m$^{-3}$), and the maximum $PM_{10}$ concentration in the Beijing area appeared at 2:00 on January 28, 2017 (up to 917 μg m$^{-3}$). This heavy pollution event swept through Beijing and the surrounding areas, reaching a maximum at almost the same time. The visibility at the corresponding time was less than 500 meters. The imaginary part of the AERI can be calculated from the visibility according to Eq. (7). Fig. 2a shows a relationship plot of the imaginary parts of the AERI and $PM_{10}$ data measured in the Beijing area; there is a strong correlation between the AERI and aerosol particle mass concentration, with a linear correlation coefficient of 0.96. The fitted line in Fig. 2a has a slope of 3845 kg m$^{-3}$. Therefore, $R_{MN}$ was taken as 3845 kg m$^{-3}$ for the Beijing area to estimate the aerosol vertical turbulent flux. Similarly, Fig. 2b shows the results for the Baoding area, and $R_{MN}$ was set to 3711 kg m$^{-3}$ for the Baoding area to estimate the aerosol vertical turbulent flux. The two ratio coefficients are relatively close. Figs. 2a and 2b also show that in the case of light pollution, Beijing's $R_{MN}$ is slightly larger.

Furthermore, Figs. 2a and 2b show that although there is a significant scattering between $PM_{10}$ and $n_{IM}$ that may be attributed to a significant separation between the two measurement locations for visibility and $PM_{10}$, there is a strong linear correlation between the imaginary part of the AERI and $PM_{10}$. The imaginary part of the AERI has a slightly stronger relationship with the $PM_{10}$ data obtained in the Baoding area than in the Beijing area.

$R_{MN}$ should be obtained by simultaneously measuring $Ma$ and the imaginary part of the AERI
at the same location with the LAS, so that real-time $R_{MN}$ can be obtained. For GC site and CAMS
site, measuring positions of $PM_{10}$ and visibility are a little far from LAS measurement. So a constant
ratio $R_{MN}$ is more representative than a simultaneous value.
The following provides the results of the aerosol turbulent flux under typical weather
conditions in Beijing and Baoding for the period from 10 March 2017 to 17 March 2017.
**4.2 Characteristics of aerosol flux in the Beijing region**
To analyze the aerosol turbulent flux characteristics, we present the time series of the
conventional meteorological parameters. The time series of temperature, RH, wind speed, wind
direction, $PM_{10}$, $C_{n,Re}^2$, $C_{n,Im}^2$ and aerosol flux are shown in Figs. 3a-3h, respectively. The
temperature has a distinct diurnal variation, indicating that this period had primarily sunny weather.
The RH from 10 March 2017 to 17 March 2017, was less than 60%, and the RH for most of the
period was less than 30%. The wind speed was low; only during the period from March 11 to March
14 was the wind moderately strong. At 6:00 on March 12, the maximum wind speed was 4.2 m s$^{-1}$.
At that time, the wind direction has diurnal variation, which is related to the sea-land breeze, valley
wind and urban heat island circulation which may exist under the control of weak weather system(Li
et al., 2019). Moreover, two light pollution events occurred (MEP, 2012) on March 11 and March
16, with $PM_{10}$ concentrations approaching 200 µgm$^{-3}$. From the data of $C_{n,Re}^2$ and $C_{n,Im}^2$ in Figs. 3f
and 3g, the real part of the AERISP $C_{n,Re}^2$ has obvious diurnal variations, i.e., smaller in the morning
and at night and larger at noon. The imaginary part of the AERISP $C_{n,Im}^2$ had no distinct diurnal
variation. According to Fig. 3g, there are some peak values, i.e., some sudden increases and
decreases, which may be related to sudden changes in wind direction, as shown in Fig. 3d.
The LAS at the CAMS site was located in the roughness layer, so the local similarity theory
should in principle applied to flux calculation. Because there was no measurement of wind speed
and temperature profiles near the LAS measurement location, the friction velocity and characteristic
temperature could not be calculated. We (Yuan et al, 2015) conducted a test experiment for aerosol
vertical flux in Hefei, China, using free convection assumptions and local similarity theories to
calculate aerosol fluxes. Comparison of the calculation results of the two methods shows that very
unstable condition accounts for about 62 % of the time, and the relative difference is about 5%.
Under weak unstable and stable condition, the relative error is about 15%.
From the aerosol flux time series given in Fig. 3h, the aerosol flux is large at noon and small
in the morning and at night, which is mainly because of the strong convection at noon. However,
large aerosol fluxes also occurred on the nights of March 11 and March 12, which were related to
high wind speeds. The mean aerosol flux measured at this observation point during this period was
0.0039 mg m$^{-2}$ s$^{-1}$.
**4.3 Characteristics of aerosol flux at the GC site**
Similarly, Figs. 4a-4d provide the time series of temperature, RH, wind speed and wind
direction at 3 meters and 18 meters for the GC site, and Figs. 4e-4h show the $PM_{10}$, $C_{n,Re}^2$, $C_{n,Im}^2$
and aerosol flux curves over time. According to Fig. 4a, the temperatures at both heights show
distinct diurnal variations. The daytime is characterized by unstable stratification, and at night,

stable stratification prevails. Moreover, in the morning and evening, there is a transition period between stable and unstable stratification. Here, $u_*$, $T_*$ and MO length $L$ were calculated from the wind speed and temperature measured at 3 m and 18 m on a meteorological tower. Fig. 4b shows a plot of the two levels of RH over time, again with apparent diurnal variations. The RH of the GC site was lower at the CAMS site. Figs. 4c and 4d provide the time series of wind speeds and wind directions at two levels. At 6:00 on March 12, the wind speed was relatively high, and the maximum at 18 meters was 6.5 m s$^{-1}$. At the same time, the maximum wind speed was reached in the Beijing area, although the speed was lower in Beijing. The overall trend of wind direction at the GC site was more consistent with the results of the CAMS site.

Figure 4e shows the PM$_{10}$ trend of over time. There were two light pollution events on March 11 and March 16. The overall trend is the same as in Fig. 3(e). Figs. 4f and 4g show the time series of the imaginary and real parts of the AERISP for the GC site. The real part of the AERISP is large at noon, and the optical turbulence is strong. The real part of the AERISP is small during the morning and evening, and the corresponding turbulence is weak. The imaginary part of the AERISP given in Fig. 4g does not show an apparent diurnal variation, and there may be some sharp peaks.

Figure 4h shows the aerosol mass vertical flux changes over time. The aerosol flux has a significant diurnal variation characteristic associated with turbulent transport near the surface. The mean aerosol flux measured at the GC site during this period was 0.0016 mg m$^{-2}$ s$^{-1}$. This value is much smaller than the results for the CAMS site. Human activities contribute to increased water vapor releases in urban areas compared to rural areas, as observed by Dou et al ( 2014), and especially for the night-time SBL in winter. During our observation period, the RH of the city was lower than the rural area. However, human activities cause more aerosol particles in urban areas than in rural areas.

**4.4 Aerosol flux during HPEs**

In the winter of 2016, there were several HPEs. A heavy pollution event began on 1 December 2016 and ended on 10 January 2017. Relevant observational experiments were performed in the Beijing and Baoding areas, including observations of meteorological parameters and aerosol parameters, to understand the causes of the heavy pollution.

According to the definition of HPEs and classification, there were 7 TS stages in the 2016 winter heavy pollution event, and the AS stage appeared immediately after 4 TS stages. These included 00:00 on December 1 to 03:20 on December 4, 18:40 on December 15 to 00:00 on December 22, 00:00 on December 29 to January 2, and 00:00 and 8:40 on January 2 to 00:00 on January 5.

During this period, we used a LAS to conduct an observational study of the vertical aerosol flux in the GC site, which was from 00:00 on December 1, 2016, to 00:00 on December 22, 2016. No corresponding observations were made at the Beijing site during this period. Here, we first discuss the observation results of the GC site, Baoding City, as shown in Fig. 5. Fig. 5a shows the time series of the aerosol vertical turbulent flux. Figs. 5b-5g indicate the time series for the real and imaginary parts of the AERISP, the temperature and RH at 18 meters, and the wind speed and direction. Purple curves indicate the TS stages, red curves show the AS stages, and grey curves show the RS stages.

According to Fig. 5a, in the TS stages and the RS stages, the aerosol flux exhibited diurnal

variations, while the AS stage did not show a diurnal variation. There were some peaks in the TS stage. The average aerosol flux of the TS stages was 0.00065 mg m$^{-2}$ s$^{-1}$, the average value of the AS stages was 0.00025 mg m$^{-2}$ s$^{-1}$, and the average value of the RS stages was 0.00063 mg m$^{-2}$ s$^{-1}$. The aerosol turbulent fluxes in the TS and RS stages were similar, while the aerosol turbulent flux in the AS stage was much smaller than the TS and RS stages.

According to Fig. 5b-5c, the imaginary structure parameters and the real structure parameters of the refractive index in the TS and RS stages exhibited diurnal variations, while the AS stage did not exhibit a diurnal variation. Fig. 5d shows that except for the second AS event (22:00 on 19 December 2016 to 00:00, 22 December 2016), the temperature showed a diurnal variation. During the AS stage, the RH (see Fig. 5e) was close to 100%, while the RH during the TS and RS stages were lower. Moreover, Fig. 5f shows that during this time, the wind speed was relatively weak, although the wind speed was slightly stronger on December 5. As shown in Fig. 5g, during the TS and AS stages, southerly winds prevailed, while during the RS period, northerly winds prevailed. The high wind speed and convection in the TS and RS stages contributed to the upward transport of aerosol particles, whereas the low wind speed and stable stratification in the AS stage were not conducive to the upward transport of aerosol particles.

During the heavy pollution period from 1 December 2016 to 10 January 2017, we did not conduct surface aerosol flux observations at the CAMS site. From January 25 to January 31, the pollution in the Beijing area also reached the level of heavy pollution. During this HPE, a measurement of surface aerosol fluxes at the CAMS site was conducted. Figure 6 shows the results of the meteorological and pollutant observations for six days from 00:00 on January 25, 2017 to 00:00 on January 31, 2017. According to Fig. 6, northerly winds prevailed after noon on January 26, when the concentration of PM$_{10}$ dropped rapidly from 254 μgm$^{-3}$ at 12:00 to 5 μgm$^{-3}$ at 15:00. During the period 12:00-24:00 on January 26, the average wind speed was 2.6 ms$^{-1}$. On January 27, southerly winds prevailed, the average wind speed was only 0.8 ms$^{-1}$, and the aerosol concentration (PM$_{10}$) increased slowly; the increase began at 6:30 before growing rapidly at 17:50, reaching more than 300 μgm$^{-3}$ at 23:00 and 917 μgm$^{-3}$ at 2:00 am on January 28, which was the maximum aerosol concentration over the 6 day period. Then, the aerosol concentration decreased gradually. The average wind speed on January 27 was 0.6 ms$^{-1}$, southerly winds prevailed, and the mean PM$_{10}$ concentration was 440 μgm$^{-3}$, which constitutes a severe pollution level. The average PM$_{10}$ concentration during the period from 00:00 on January 25 to 00:00 on January 31 was 170 μgm$^{-3}$.

According to the previous characteristics for the TS and AS stages, a period of southerly winds can be determined as the TS stage. Thus, January 27 can be designated as the TS stage, January 28 can be determined as the AS stage, and January 29 can be determined as the RS stage. During Beijing's heavy pollution event in January 2017 (20170125-20170131), the mean aerosol vertical flux in the TS stage was 0.0024 mg m$^{-2}$ s$^{-1}$, the average value during the AS stage was 0.00087 mg m$^{-2}$ s$^{-1}$ and the RS stage was 0.0049 mg m$^{-2}$ s$^{-1}$. The overall average value was 0.0032 mg m$^{-2}$ s$^{-1}$.

Even during heavy pollution events, the RH in Beijing was lower than in the outer suburbs. According to Fig. 6e, the RH exceeded 60% in the period from 3:00 to 6:00 on January 26, where the maximum value was 63%, and the RH was less than 60% in the remaining periods. In urban areas, when the RH is low, heavy pollution incidents can occur. In Beijing, during the AS stage, the vertical flux of aerosol was less than during the TS and RS stages.

# 5 Discussions and conclusions

During the winter of 2016 and the spring of 2017, HPEs frequently occurred in the BTH area. This study investigated the aerosol vertical mass flux and compared its magnitude during different stages of HPEs, including RSs, TSs, and ASs, in two representative urban and rural sites, including the CAMS site in Beijing and the GC site in Hebei Province. Based on the light propagation theory and surface-layer similarity theory, the aerosol vertical mass flux was obtained by combining LAS observations, surface $PM_{2.5}$ and $PM_{10}$ mass concentrations, and meteorological observations, including air temperature and RH. We found that under favorable meteorological conditions for pollution dispersion, i.e., from 10 March 2017 to 17 March 2017, the vertical aerosol mass flux exhibited striking diurnal variations, with the mass fluxes reaching peak values at noon and lowering in the morning and evening. During the HPEs from 25 January 2017 to 31 January 2017 in Beijing, the vertical aerosol mass flux varied substantially during the different stages. Specifically, the mean mass flux decreased by 51% from 0.0049 mg $m^{-2}s^{-1}$ in the RSs to 0.0024 mg $m^{-2}s^{-1}$ in the TSs, which was partly due to the wind speed reduction from strong northerly winds in the RSs to southerly winds in the TSs. During the ASs, the mean mass flux decreased further to 0.00087 mg $m^{-2}s^{-1}$, which accounted for approximately 1/3 of the flux during the TSs. The weakened mass flux would further facilitate aerosol accumulation. During the HPE from December 01, 2016, to December 22, 2016, in Gucheng, the mean mass flux was similar in the RSs and TSs, ranging from 0.00063 mg $m^{-2}s^{-1}$ to 0.00065 mg $m^{-2}\,s^{-1}$. This is partly because Gucheng was less affected by strong northerly winds than Beijing. Thus, the wind speed varied slightly from the RSs to TSs. However, the mass flux decreased substantially to 0.00025 mg $m^{-2}s^{-1}$ in the ASs, which was merely 1/3 of the mean flux in the TSs.

Based on our measurement results, it can be seen that from the TS to the AS, the aerosol vertical turbulent flux decreased, but the aerosol particle concentration with surface layer increased. it is inferred that in addition to the contribution of regional transport from upwind areas during the TS, suppression of vertical turbulence mixing confining aerosols to a shallow boundary layer increased accumulation.

In this study, the aerosol emission flux was also estimated in these two rural and urban sites. Generally, compared with the emissions in spring, we found that in winter, the near-ground emissions were weaker in suburban areas and were similar in urban areas. In suburban areas, although the aerosol concentrations were relatively high (Shen et al., 2018), the upward emitted aerosol flux was smaller than in urban areas. During the ASs of the HPEs, the aerosol emission flux from near-ground emission sources was weaker than for the RSs and TSs at both the CAMS and GC sites, which indicates that surface pollutant emissions are not the major cause of explosive $PM_{2.5}$ growth. During the ASs with weak solar radiation, the factors most associated with aerosol concentration changes were horizontal transport and BL height variations, which might be the leading causes of increased $PM_{2.5}$. This is in line with previous studies that the main reason for the explosive growth of aerosol concentration during AS is attributed to the horizontal transport during TS. The TS will definitely appear before CS. The south or southwest wind will always appear in the TS, and the concentration of PM10 in Baoding is higher than the mass of $PM_{10}$ in Beijing, which is generally maintained for one to two days. Except for the southerly or southwesterly winds for one to two days, there will be no CS in Beijing. Even if it is a southerly or southwesterly wind, if the wind speed is too small ($<1ms^{-1}$), AS will not appear. Only the southerly or southwesterly wind with a wind speed greater than a specific value ($>1.5$ m $s^{-1}$), and the concentration of $PM_{10}$ in the area to

the south of Beijing is higher than that in Beijing, and then there will be CS after a small wind
(Zhong et al., 2018c; Zhong et al., 2018b; Zhang et al., 2018).
Compared to the results (Yuan et al. 2016) from Hefei, China, a small and medium-sized
provincial capital city in East China, the measured aerosol mass-fluxes in Beijing are almost at the
same amount. A series of measures and actions have been made for emission reduction in Beijing,
and the main emission is from vehicles. The difference in aerosol mass flux may be small.
Due to the lack of necessary experimental conditions, such as meteorological towers and EC
systems, current experimental results cannot be compared with EC methods. According to the
literature data, the two methods have been compared indirectly, and the estimated aerosol flux under
different measurement conditions is consistent in magnitude (Yuan et al., 2016). However, a direct
comparison of the two methods is in development.
Compared with the EC method, the aerosol flux has high spatial representativeness based on
the principle of light propagation, and there is no need to install a tall tower. However, the estimation
of aerosol fluxes using the LAS method still has theoretical and practical deficiencies. At present,
the LAS method for the aerosol flux regards the aerosol particles as ordinary scalar molecules. At
the same time, based on the assumption of the equivalent medium, the imaginary part of the AERI
is taken for granted as proportional to the aerosol mass concentration. This is often not the case. The
actual turbulence spectrum shape may deviate from von Karman spectrum, and turbulence
intermittent and scintillation saturation can also occur (Moene et al., 2009). The applicability of the
near-surface layer similarity theory to the aerosol particle motion under stable layer conditions also
has many problems. The formation of new particles and changes in aerosol particle size distribution
also affect the scintillation in light propagation. There are also practical problems such as untimely
maintenance, rainfall and low visibility, and platform vibrations required for observation. All these
problems will cause errors in final estimates, so more theoretical and experimental research is
needed.
**Data availability.** Requests for data that support the findings of this study can be sent to
rmyuan@ustc.edu.cn.
**Competing interests.** The authors declare that they have no conflict of interest.
**Author contributions**. Renmin Yuan and Xiaoye Zhang designed experiments and wrote the manuscript;
Renmin Yuan, Hao Liu, Yu Gui, Bohao Shao, Yaqiang Wang, Junting Zhong and Xaioping Tao
carried out experiments; Renmin Yuan analyzed experimental results. Yubin Li and Zhiqiu Gao
designed experiments and discussed the results.
**Acknowledgements**. This study was supported by the National Key Research and Development Program
under grant no. 2016YFC0203306 and the National Natural Science Foundation of China
(41775014, 51677175). We also thank two anonymous reviewers for their constructive and helpful
comments.

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

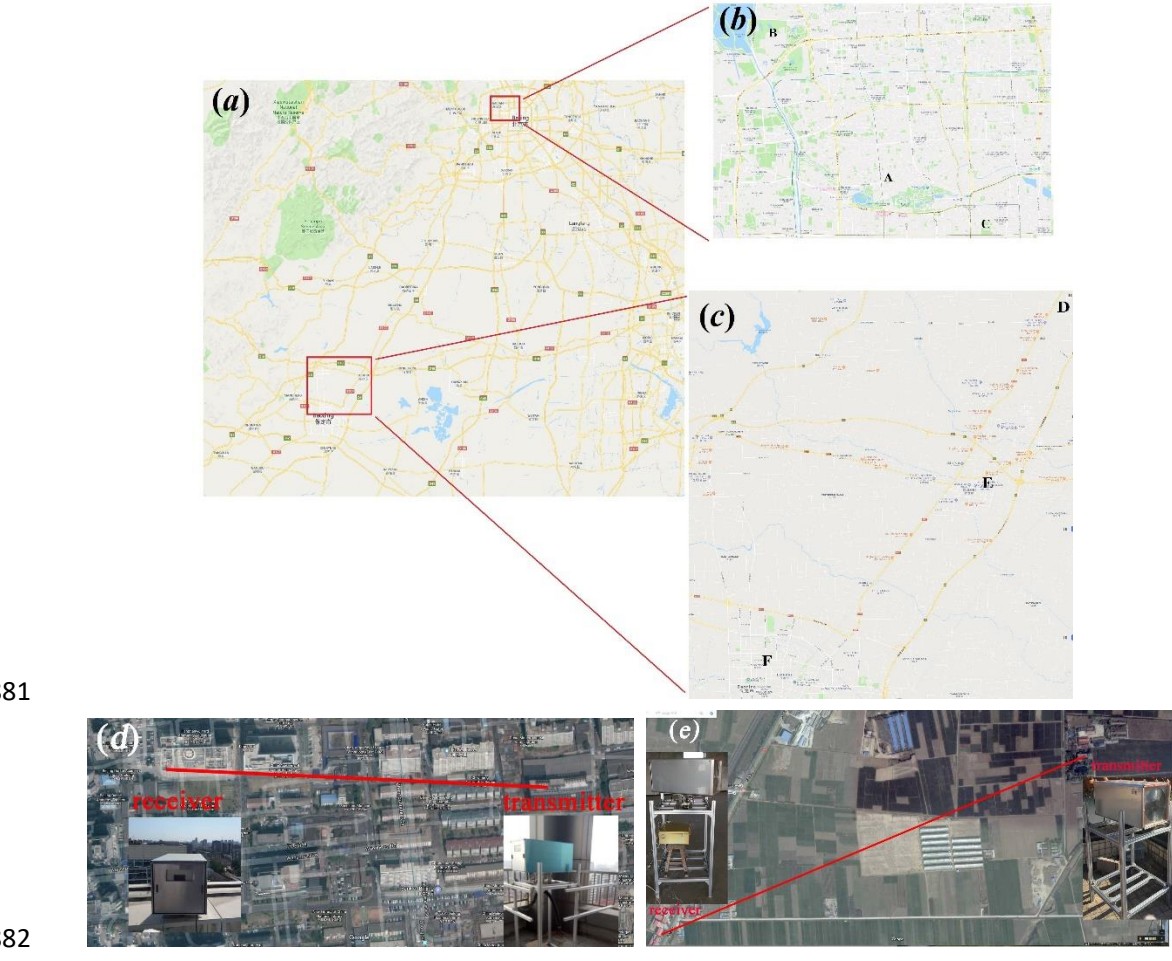



Figure 1. Photographs of the measurement site. (a) Map of the experiment area in the Beijing urban area
and suburban area and (b) expanded view of the Beijing experiment area, which is marked as the
rectangle in (a). (c) Expanded view of the Baoding experiment area, which is marked as the rectangle in
(a). (d) Satellite image of the CAMS site and (e) the satellite image of the GC site. Figs. 1a, b, c, and d
© Google.

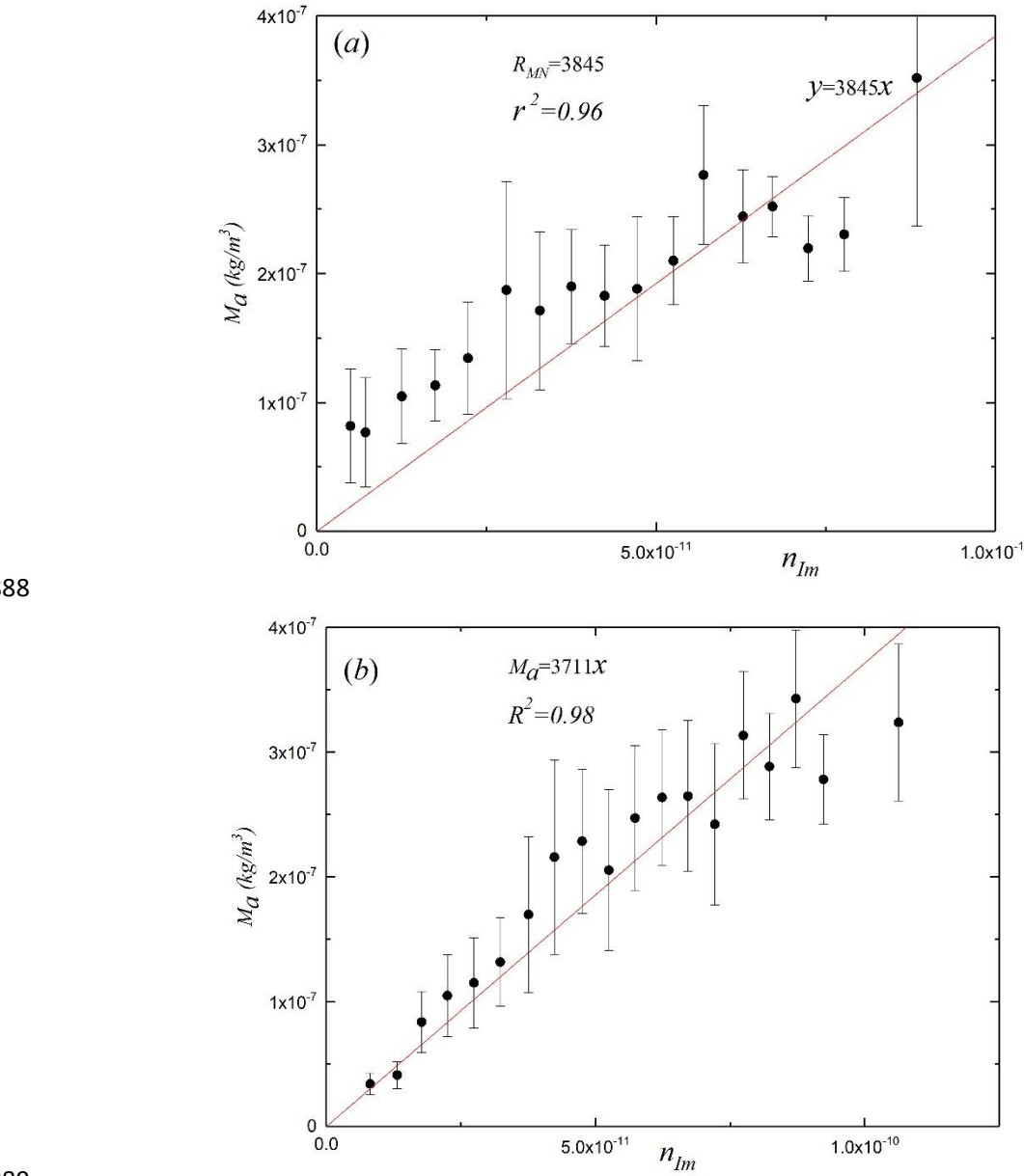




Figure 2. Relationship plot of aerosol mass concentration $M_a$ and the imaginary part of the AERI for (a)
the Beijing area and (b) the Baoding area.

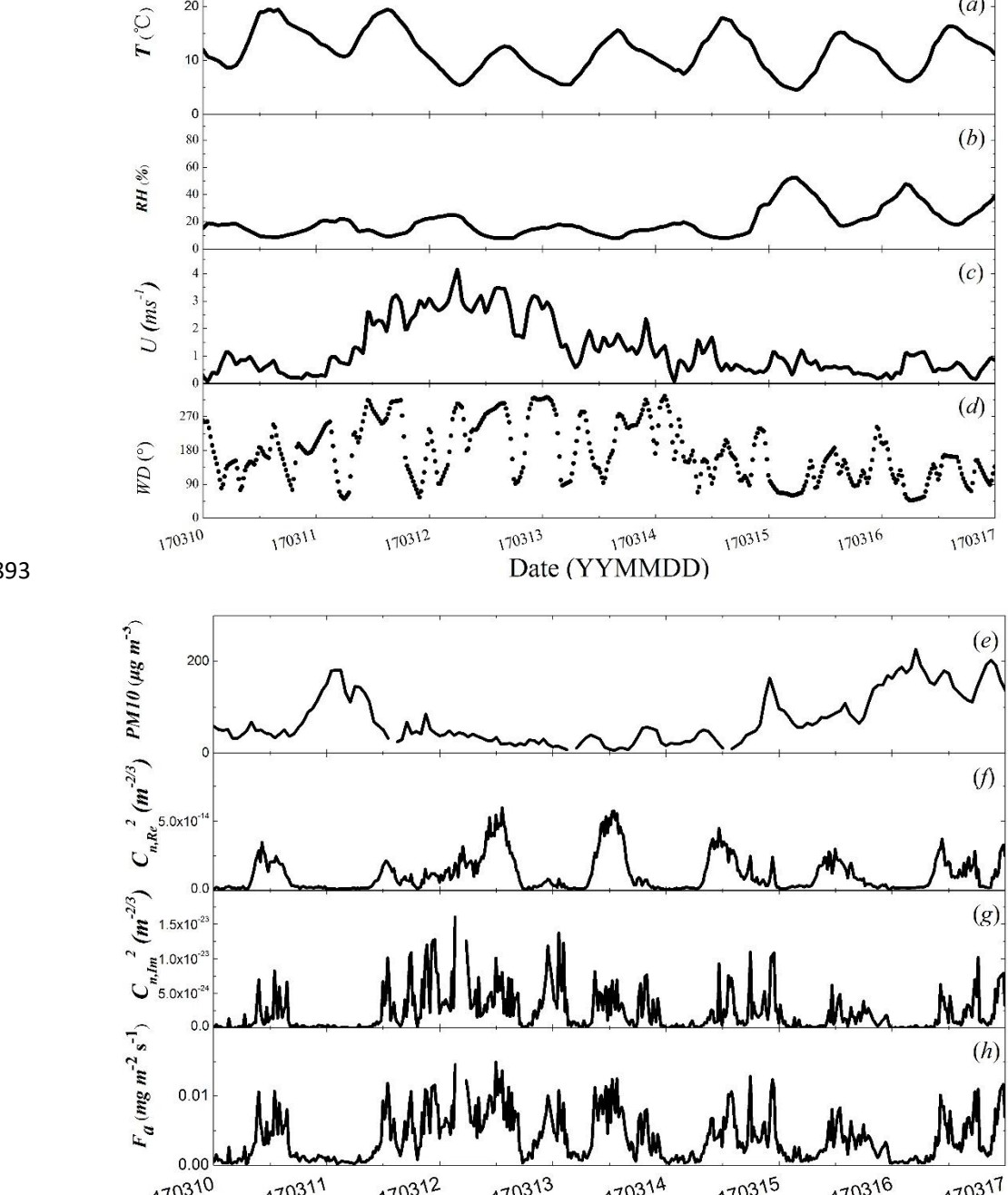



Figure 3. Temporal variations in (a) air temperature, (b) RH, (c) wind speed, (d) wind direction, (e) PM$_{10}$,
(f) real part of the AERISP, (g) imaginary part of the AERISP and (h) aerosol mass flux in the Beijing
area from March 10, 2017 to March 17, 2017.

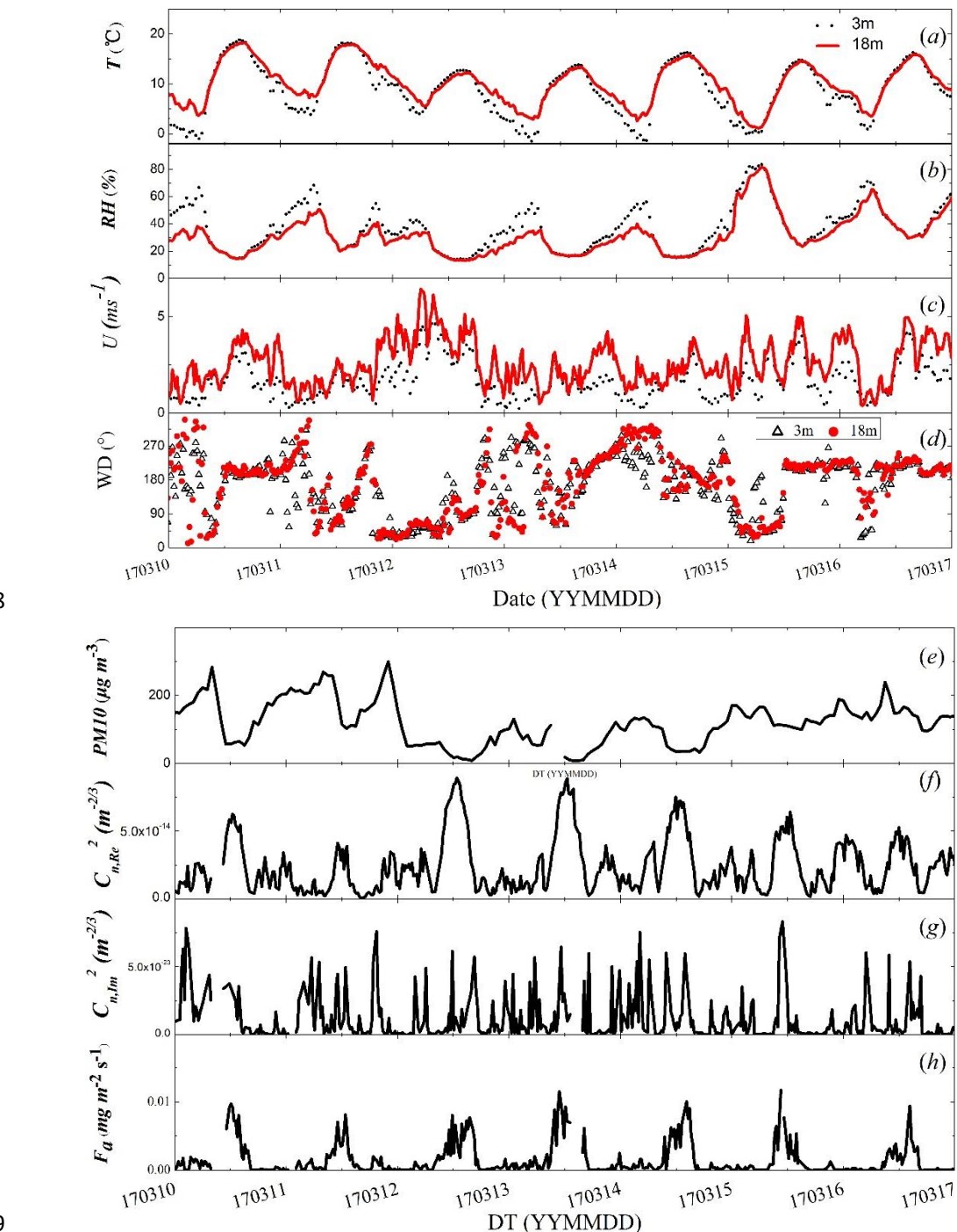



Figure 4. Temporal variations in (a) air temperature, (b) RH, (c) wind speed, (d) wind direction, (e) PM$_{10}$,
(f) real part of the AERISP, (g) imaginary part of the AERISP and (h) aerosol mass flux in the Baoding
area from March 10, 2017 to March 17, 2017.

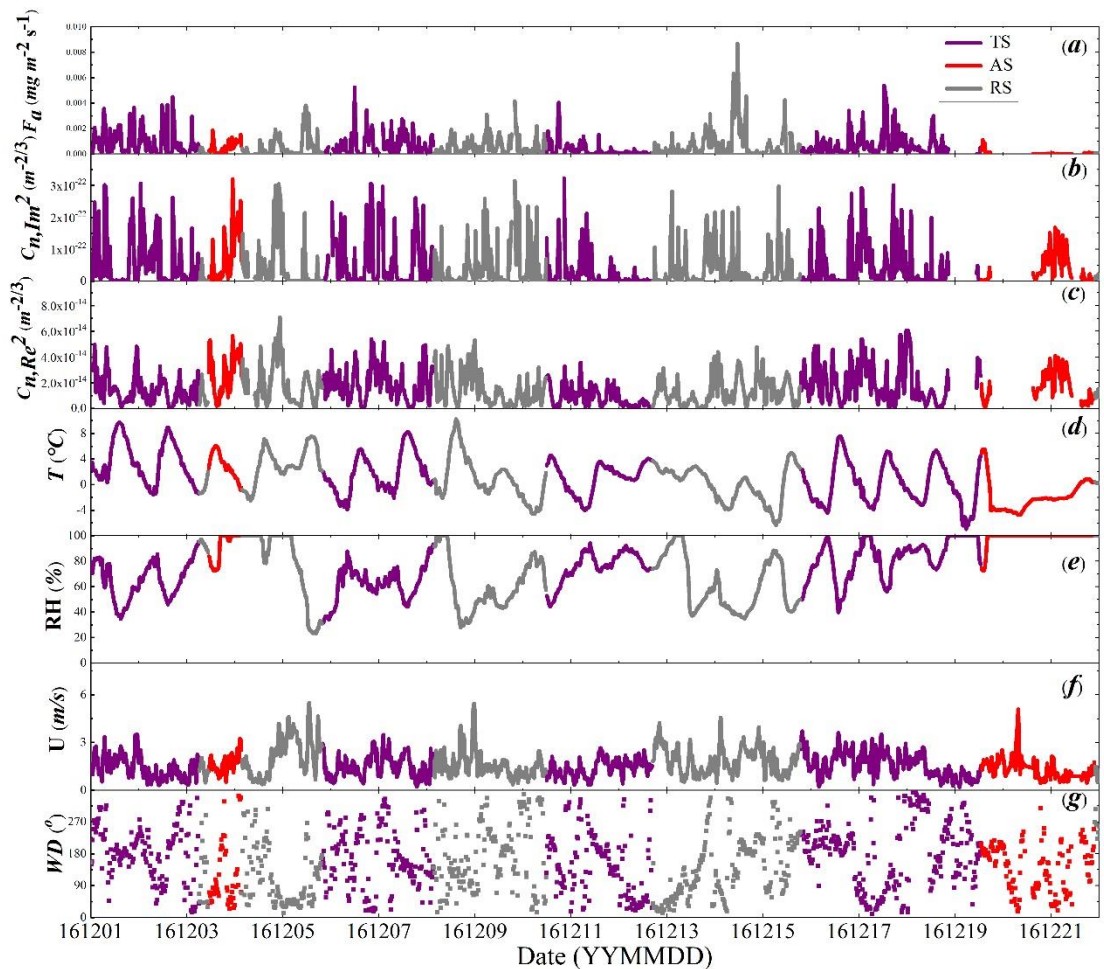


Figure 5. Temporal variations in (a) aerosol flux, (b) imaginary part of the AERISP, (c) real part of the
AERISP (d) air temperature, (e) RH, (f) wind speed, and (g) wind direction in the Baoding area during a
heavy pollution period, i.e., December 1, 2016 to December 22, 2016.

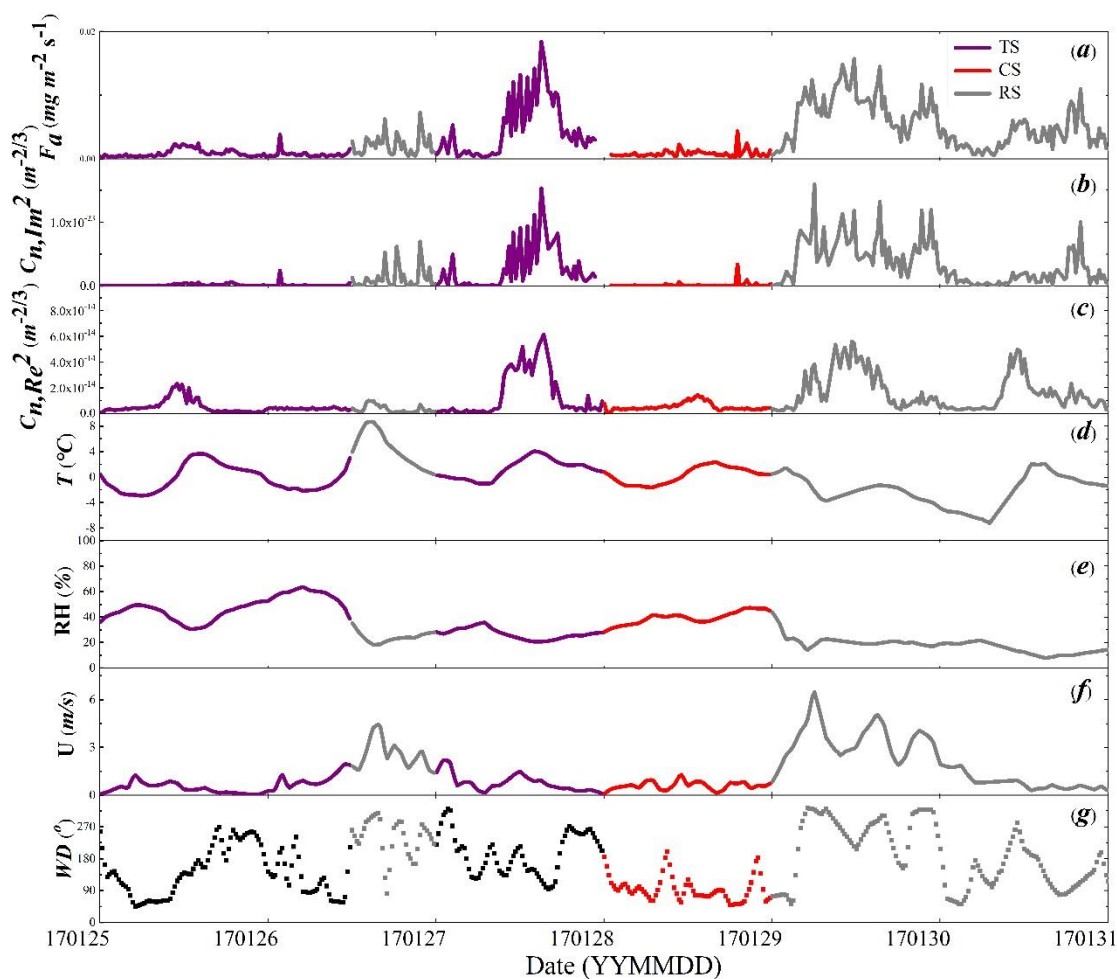


Figure 6. Temporal variations in (a) aerosol flux, (b) imaginary part of the AERISP, (c) real part of the
AERISP (d) air temperature, (e) RH, (f) wind speed, and (g) wind direction in the Beijing area
during a heavy pollution period, i.e., January 25, 2017 to January 31, 2017.
