# Peer review of "Aerosol Vertical Mass Flux Measurements During Heavy Aerosol Pollution Episodes at a Rural Site and an Urban Site in the Beijing Area of the North China Plain"

_Atmospheric Chemistry and Physics, 2018_

## Referee Comment (RC1) · Anonymous Referee #2 · 3 May 2019

General comments

Quantification of the aerosol mass flux is an important topic to understand pollutant emissions and transport over areas exposed to pollution episodes. The study utilizes an innovative large aperture scintillometer (LAS) technique to estimate the transport of aerosols over extended areas. The presented results are a valuable contribution to the understanding the emissions in urban areas and rural polluted regions.

However, since the LAS technique is semi-empirical, then additional information on

testing and evaluation of such measurements would help to improve confidence in results and understand the underlying uncertainties. For example, the LAS technique is capable to determine the magnitude of the flux but not the sign. In general the aerosols are very heterogeneous in space and the measured fluxes show typically large variation in magnitude including the sign. Over the polluted areas, which behave as the source, the emissions presumable overwhelmingly exceed the deposition sinks. Therefore, for example, a rough quantification of the deposition sink would allow to conclude that the sink term is indeed negligible and the flux quantified by LAS can be safely assumed to represent the upward fluxes. If available, the reference to comparison of the LAS method results with a more direct micrometeorological measurement would be very useful (if this was done in Yuan et al., 2016, please mention explicitly).

The manuscript would benefit also from better improved description/definition of the heavy pollution episodes (HPEs), how they are divided into stages of transport (transport stage TS), a cumulative stage (CS) and a removal stage (RS), and in particular what are the prevailing meteorological and aerosol emission/transport conditions during such episodes. This would help readers who are not familiar with HPE mechanisms more easily to follow the manuscript. According the author the TS is the period when the pollution over the measurement location was mainly contributed by the downwind pollution sources. But presumable also the local sources were also a significant contribution because the aerosol fluxes did not differ much in magnitude from subsequent phases. The CS (perhaps would be better to call accumulation stage?) represents the period of rapid accumulation of pollutants and it is not evident of this occurs because of downwind transport of pollutants trapped in the atmospheric boundary layer or local emissions or both. Therefore, it is not clear if the stage differs from the TS in terms of location of emission sources or difference is made by the meteorological conditions favouring accumulation of the pollutants in the ABL. Regarding the RS, presumably the pollutant concentrations drop due to the atmospheric mixing and transport to higher levels. The other possibility is removal by scavenging or dry deposition. Dry deposition however is a slow process and also the results do not support such assumption (up-

ward fluxes in Figs. 5 and 6 during the RS). The explanation in l. 425 is confusing as if the particles are removed from the atmosphere and reduction in pollutants does not occur because of the atmospheric mixing (and upward transport of aerosols). In relation to interaction between the aerosol pollution and meteorology, the authors suggest in the abstract (and l. 498-500) that the aerosol pollution had an effect to turbulence intensity leading to further weakening of mixing and increased accumulation. Such effect is not directly evidenced by the results in the manuscript (or cannot be distinguished) and should be further supported by the literature references rather than stated as the result.

The manuscript would benefit also from numerous minor improvements and language editing. Please see my specific comments below.

Specific comments

1. Line 28-29, sentence difficult to follow, please revise.

2. L. 35-36, the statement is vague, see also general comments.

3. L. 60-61 "the consumption of a product" – revise phrasing

4. L. 77-79: the EC method has been used already for decades to quantify the aerosol particle number fluxes. As an example of earlier studies, see e.g. Buzorius, G., Rannik, Ü., Mäkelä, J.M., Vesala, T., Kulmala, M., 1998. Vertical Aerosol particle fluxes measured by eddy covariance technique using condensational particle counter. J. Aerosol Sci., 29, 157-171.

5. L. 80, The EC method enables to determine the vertical turbulent flux, which can be different from total vertical transport. Also, the flux is provided by the cross-covariance (and not correlation).

6. L. 82-83, the EC principle allows to quantify the number flux from fluctuation measurements, rephrase the sentence.

[Figure]

7. L. 105, The eddy correlation principles have been widely used (or something like this, revise the sentence)

8. L. 126 "how much the surface emissions contribute to the concentration of pollutants"

9. L. 142-143, phrasing is not good. Rather the transport properties or the statistical aerosol transport is similar to that of scalars? In fine detail the aerosol motion can be different from the air motion and the statement is not strictly correct.

10. L. 166-167, temperature is not a passive atmospheric constituent because buoyancy affects strongly the motion of air. Also "distribution" does not seem relevant but maybe just "small particles". Rather say that similarity of atmospheric aerosols and temperature can be assumed for the purpose.

11. L. 173 "aerosol particles are continuously dispersed in the air", the meaning and purpose of this sentence is not clear.

12. L. 192, Correct $R_{MN}$

13. L. 209-212, please provide reference and/or explanation for the relation between the high/low frequency fluctuations and the real/imaginary parts of the AERI.

14. L. 225, turbulent fluctuations of what?

15. L. 297, e.g. stands for "for example", not relevant here.

16. L. 309-310. The method for judging.. sentence difficult to follow, rephrase.

17. L. 315, how was "mean of the adjacent difference" defined, based on the moving average or how? Improve wording of the sentence.

18. L. 321-321, is the exact shape of the spectrum relevant? Or the method relies purely on the Kolmogorov's power laws of the spectra?

19. L. 328, "heavy pollution weather conditions" is a bit weird, please rephrase

20. L. 361, rather the wind direction varied throughout day?

21. L. 368-374. The "free convection" conditions are not always easily satisfied. Free convection means that the buoyancy-driven turbulence dominates over mechanical turbulence and this is not just the unstable conditions but the free convective limit of the unstable conditions. Please clarify and evaluate the uncertainties introduced by such assumption.

22. L. 421, southerly wind conditions

23. Figures 3-6, the square value of the structure parameter is plotted according to label in y-axis of the relevant subplots.

24. Discussion and conclusions: how do the measured aerosol mass-fluxes compare with relevant literature values and/or earlier measurements and typical emission estimates? Please discuss this how to results contribute to understanding of pollution emissions.

---

## Referee Comment (RC2) · Anonymous Referee #1 · 28 May 2019

East Asia, especially China, is facing heavy haze pollution in wintertime. Though many measurements on air pollutants have been extensively conducted across China, there is still a lack of flux data on pollutants, which may play a substantial role in haze formation. This study combines measurements on meteorological conditions, flux, as well as PM2.5 concentration in BTH region to derive aerosol vertical mass flux, and provide some observational insight on aerosol vertical flux under stable condition. Therefore, this manuscript adds to our current knowledge of aerosol vertical exchange and its impact on meteorology. However,I have some concerns about the methods/data analysis

used in the study and the interpretation of results , and more in-depth analysis and discussion ought to be provided.I think this manuscript can be considered for publication only if the authors could adequately address the comments below.

Major comment: 1.There are two observation sites, a rural site (GC site) and an urban site (CAMS site). The Monin-Obukhov similarity theory (MOST) is applied in rural site because the surface is homogenous. But in the urban site, the observation was within the urban roughness sublayer (3-5 mean building height), MOST is invalid due to the lack of constant-flux conditions, the local similarity theory should be used. In other words, the function or the parameters in the similarity relationship should be different for the rural and urban site.

2.The function and parameters of the similarity relationship are not universal, the authors should explain why they use these function and parameters in the paper. For example, in Eq. 4, the authors said that they take the parameters b1 and b2 follow DeBruin et al., 1995. But in DeBruin et al., 1995, it said that "For stable conditions there is no consensus on the universal function", b1=5,b2=0 were found by DeBruin et al., 1993, and "the scatter was very large". So DeBruin may not be the best choice. Especially, in Yuan et al., 2016, the parameter b1 and b2 follows Wyngaard et al., 1971., which is very different from DeBruin et al., 1993. When b1 and b2 follow DeBruin et al., 1995, it means that $\eta(\xi)$ stays constant with stability; but when b1 and b2 follow Wyngaard et al., 1971, it means that $\eta(\xi)$ changes constant with stability. The author should explain why they choose DeBruin et al., 1995.

3.In L359 The conventional meteorological parameters were measured at 20m above the ground surface. But in L275, the author said that the measurement heights of temperature and wind speed were 1.5 m and 10 m at CAMS site (Beijing). It should be clear which data were used to calculate the aerosol flux. Because the average height of the building was 24m in CAMS site, and LAS was located at 43meters. The temperature measured at 1.5m within the canopy layer is different from 43m above the canopy layer, and the calculation of aerosol fluxes from Eq. 12 was badly influenced.

[Figure]

4. Another issue that the authors need to address is the assumption between AERI( atmospheric equivalent refractive index) and aerosol mass concentration as well as aerosol adsorption. First, there do exist some light-absorbing trace gases in the atmosphere, which may influence AERI significantly. Second, aerosol absorption generally contribute a relatively small part of the extinction. By contrast, scattering components like sulfate and organic matters dominate aerosol extinction during haze pollution episode, especially under high humidity. Last but not at least, aerosol extinction is also closely related to the number concentration and size distribution, which need to be considered here. I do not think it is technically robust to simply get the relationship between the imaginary part of the AERI and the atmospheric aerosol mass concentration in Eq.6.

Minor issues: Some statements in this manuscript are very hard to follow. Language editing is needed for improving the accuracy of language as well as overall readability.

Line 43: Please rephrase 'heavy pollution weather'

Line 48: 'few studies' should be 'few study'

Line 72: what is the boundary layer box model? Usually box model is zero-dimensional.

Line 106: should be 'makes it possible'

Eq. 11: replace z with (z-d)

Line 304:More detail needed, not "personal experience".

Line 378: weakly unstable is not free convection. The free convection assumption was not satisfied at night.
* * *

---

## Author Comment (AC1) · 29 Jun 2019

Authors reply to reviewer's comments:

Dear Anonymous Referee,

Thanks for your careful review of the manuscript. We read the reviewer's comments carefully, and have responded and taken all of the reviewer's comments into consideration and revised the manuscript accordingly. My detailed responses are as follows: Comments from Anonymous Referee #2: "Quantification of the aerosol mass flux is an

important topic to understand pollutant emissions and transport over areas exposed to pollution episodes. The study utilizes an innovative large aperture scintillometer (LAS) technique to estimate the transport of aerosols over extended areas. The presented results are a valuable contribution to the understanding the emissions in urban areas and rural polluted regions."

1) "However, since the LAS technique is semi-empirical, then additional information on testing and evaluation of such measurements would help to improve confidence in results and understand the underlying uncertainties. For example, the LAS technique is capable to determine the magnitude of the flux but not the sign. In general the aerosols are very heterogeneous in space and the measured fluxes show typically large variation in magnitude including the sign. Over the polluted areas, which behave as the source, the emissions presumable overwhelmingly exceed the deposition sinks. Therefore, for example, a rough quantification of the deposition sink would allow to conclude that the sink term is indeed negligible and the flux quantified by LAS can be safely assumed to represent the upward fluxes. If available, the reference to comparison of the LAS method results with a more direct micrometeorological measurement would be very useful (if this was done in Yuan et al., 2016, please mention explicitly)."

Response: Thanks for your suggestion. We have added the statement that the sink term is indeed negligible and the flux quantified by LAS can be assumed to represent the upward fluxes. Please see L288-290. At present, we have not conducted more direct meteorological measurements to obtain aerosol fluxes, such as the use of EC methods for aerosol flux measurements. Next, we will compare the aerosol flux obtained by LAS with the aerosol flux received by the EC method.

2) "The manuscript would benefit also from better improved description/definition of the heavy pollution episodes (HPEs), how they are divided into stages of transport (transport stage TS), a cumulative stage (CS) and a removal stage (RS), and in particular what are the prevailing meteorological and aerosol emission/transport conditions during such episodes. This would help readers who are not familiar with HPE mechanisms

more easily to follow the manuscript."

Response: Based on meteorological causes of the increase or decrease in PM2.5 mass concentrations, the HPEs are divided into TSs, ASs (in the new version, CS denoted as AS by suggestion), and RSs. During the TSs, the PM2.5 is dominated by relatively strong southerly winds, which carry polluted air masses from more populated southern industrial regions (Guo et al., 2014; Zhong et al., 2018a). Before rising processes during TSs, the urban PM2.5 mass concentration of Baoding, which is typically representative of pollution conditions in the south of Beijing, was much higher than Beijing; the winds in Beijing rapidly shifted from northerly to southerly. Then the rising in PM2.5 occurred, consistently with southerly slight or gentle breezes in the BL. The southerly air mass moves more than 288 km d-1 below 500 m (estimated from the measured wind speed), which are fast enough to transport pollutants to Beijing. Such processes indicate southerly pollutant transport is primarily responsible for the rising, given the pollution transport pathway of the southwest wind belt determined by the unique geographic features of the North China Plain, with the Tai-hang Mountains and the Yan Mountains strengthening the southwest wind belt and leading the convergence of pollutant transport in Beijing (Su et al., 2004). During the ASs, PM2.5 increase is dominated by stable atmospheric stratification characteristic of southerly slight or calm winds, near-ground anomalous inversion, and moisture accumulation. When the vertical aerosols are accumulated to a certain degree, the dominant scattering aerosols will substantially back-scatter solar radiation, causing a reduction in the amount of solar radiation that reaches the surface, which creates a near-ground cooling effect through atmospheric circulation and vertical mixing(Zhong et al., 2018b). The temperature reduction induces or reinforces an inversion that further weakens turbulence diffusion and results in a lower BL height, which further worsens aerosol pollution. This condition also decreases the near-ground saturation vapor pressure and suppresses water vapor diffusion to increase the relative humidity (RH), which will further enhances aerosol hygroscopic growth and accelerates liquid-phase and heterogeneous reactions to worsen aerosol pollution (Ervens et al., 2011; Kuang et al., 2016; Pilinis et al., 1989;

[Figure]

Zhong et al., 2018a; Zhong et al., 2018b). This feedback effect of further worsened meteorological conditions aggravates PM2.5 pollution (Zhong et al., 2017). During the RSs, strong northwesterly winds whose velocity increases with height occur mostly. Strong northerly winds are from less populated north mountainous areas and carry unpolluted air masses to Beijing, which is favorable for pollution dispersion. We have added some descriptions about three stages in the introduction. Please L135-156 in Section introduction. Ervens, B., Turpin, B.J., Weber, R.J., Secondary organic aerosol formation in cloud droplets and aqueous particles (aqSOA): a review of laboratory, field and model studies, Atmos. Chem. Phys. 11(2011), 11069-11102. Guo, S. et al., Elucidating severe urban haze formation in China, Proc. Natl. Acad. Sci. U.S.A. 111(2014), 17373-17378. Kuang, Y., Zhao, C.S., Tao, J.C., Bian, Y.X., Ma, N., Impact of aerosol hygroscopic growth on the direct aerosol radiative effect in summer on North China Plain, Atmospheric Environment 147(2016), 224-233. Pilinis, C., Seinfeld, J.H., Grosjean, D., Water content of atmospheric aerosols, Atmos. Environ. 23(1989), 1601-1606. Su, F., Gao, Q., Zhang, Z., REN, Z.-h., YANG, X.-x., Transport pathways of pollutants from outside in atmosphere boundary layer, Res. Environ. Sci. 1(2004), 26-29. Zhong, J. et al., Feedback effects of boundary-layer meteorological factors on cumulative explosive growth of PM2.5 during winter heavy pollution episodes in Beijing from 2013 to 2016, Atmos. Chem. Phys. 18(2018a), 247-258. Zhong, J., Zhang, X., Wang, Y., Liu, C., Dong, Y., Heavy aerosol pollution episodes in winter Beijing enhanced by radiative cooling effects of aerosols, Atmos. Res. 209(2018b), 59-64. Zhong, J. et al., Relative contributions of boundary-layer meteorological factors to the explosive growth of PM 2.5 during the red-alert heavy pollution episodes in Beijing in December 2016, J. Meteorolog. Res. 31(2017), 809-819.

3) "According the author the TS is the period when the pollution over the measurement location was mainly contributed by the downwind pollution sources. But presumable also the local sources were also a significant contribution because the aerosol fluxes did not differ much in magnitude from subsequent phases. The CS (perhaps would be better to call accumulation stage?) represents the period of rapid accumulation of

pollutants and it is not evident of this occurs because of downwind transport of pollutants trapped in the atmospheric boundary layer or local emissions or both. Therefore, it is not clear if the stage differs from the TS in terms of location of emission sources or difference is made by the meteorological conditions favouring accumulation of the pollutants in the ABL. Regarding the RS, presumably the pollutant concentrations drop due to the atmospheric mixing and transport to higher levels. The other possibility is removal by scavenging or dry deposition. Dry deposition however is a slow process and also the results do not support such assumption (up-ward fluxes in Figs. 5 and 6 during the RS). "

Response: After a series of measures and actions, including air pollutant emission reduction, energy structure adjustments to decrease the dependence on fossil fuels, and other supportive policies, the emission sources in Beijing are strikingly less than the polluted southern industrial regions with large anthropogenic emissions. Therefore, the contribution of local emissions in Beijing is relatively smaller than that in the other areas such as Baoding. The TS will appear before CS. The south or southwest wind will always appear in the TS, and the concentration of PM10 in Baoding is higher than the concentration of PM10 in Beijing, which is generally maintained for one to two days. Except for the southerly or southwesterly winds for one to two days, there will be no CS in Beijing. Even if it is a southerly or southwesterly wind, if the wind speed is too small (<1ms-1), AS will not appear. Only the southerly or southwesterly wind with a wind speed higher than a specific value (>1.5ms-1), and the concentration of PM2.5 in the area to the south of Beijing is higher than that in Beijing, and then there will be CS after a small wind. Therefore, the main reason for the explosive growth of aerosol concentration during CS is that explosive growth is attributed to the horizontal transport during TS. Please L591-603.

4) "The explanation in l. 425 is confusing as if the particles are removed from the atmosphere and reduction in pollutants does not occur because of the atmospheric mixing (and upward transport of aerosols). "

[Figure]

Response: We modified the expression. During the RSs, strong northwesterly winds whose velocity increases with height occur mostly. Strong northerly winds are from less populated north mountainous areas and carry unpolluted air masses to Beijing, which is favorable for pollution dispersion. Please L154-157.

5) "In relation to interaction between the aerosol pollution and meteorology, the authors suggest in the abstract (and l. 498-500) that the aerosol pollution had an effect to turbulence intensity leading to further weakening of mixing and increased accumulation. Such effect is not directly evidenced by the results in the manuscript (or cannot be distinguished) and should be further supported by the literature references rather than stated as the result."

Response: Based on the results in the manuscript, the effect of the aerosol pollution cannot be drawn out to turbulence intensity leading to further weakening of mixing and increased accumulation. Based on our measurement, it can be seen that from the TS to the AS, the aerosol vertical turbulent flux decreased, but the aerosol particle concentration within surface layer increased, and it is inferred that in addition to the contribution of regional transport from upwind pollution areas during the TS, suppression of vertical turbulence mixing confining aerosols to a shallower boundary layer increased accumulation. We modified some expression. Please see L580-584.

"The manuscript would benefit also from numerous minor improvements and language editing. Please see my specific comments below."

Specific comments 1) "Line 28-29, sentence difficult to follow, please revise."

Response: The sentence means a weakened turbulence intensity and low vertical aerosol fluxes in winter and polluted areas such as GC We revised. Please see L28-29.

2) "L. 35-36, the statement is vague, see also general comments."

Response: The sentence is just a reference, not a conclusion. So the sentence is

deleted.

3) "L. 60-61 "the consumption of a product" – revise phrasing"

Response: expressed as "product energy consumption". Please Line 62-63.

4) "L. 77-79: the EC method has been used already for decades to quantify the aerosol particle number fluxes. As an example of earlier studies, see e.g. Buzorius, G., Rannik, Ü., Mäkelä, J.M., Vesala, T., Kulmala, M., 1998. Vertical Aerosol particle fluxes measured by eddy covariance technique using condensational particle counter. J. Aerosol Sci., 29, 157-171."

Response: The manuscript was revised according to the comment. Please see line 86-87.

5) "L. 80, The EC method enables to determine the vertical turbulent flux, which can be different from total vertical transport. Also, the flux is provided by the cross-covariance (and not correlation)."

Response: "eddy correlation" modified to "eddy covariance"

6) "L. 82-83, the EC principle allows to quantify the number flux from fluctuation measurements, rephrase the sentence."

Response: Based on EC principle, the vertical velocity fluctuations and the fluctuations in the aerosol particle number density can be measured, and the EC principle allows to quantify the number flux from fluctuation measurements. We modified the sentences. Please see Line 88-89.

7) "L. 105, The eddy correlation principles have been widely used (or something likethis, revise the sentence)."

Response: "eddy correlation" modified to "eddy covariance."

8) "L. 126 how much the surface emissions contribute to the concentration of pollutants"

Response: At present, the question cannot be answered. We hope that with the measurement of the near-surface aerosol vertical flux, the results will help understand the accumulation of local aerosol concentrations, how much the surface emissions contribute to the level of pollutants, and how much the concentration of pollutants is attributed to upwind areas. This article does not discuss this issue for the time being. We will combine the data from the lidar and present it in a later article.

9) "L. 142-143, phrasing is not good. Rather the transport properties or the statistical aerosol transport is similar to that of scalars? In fine detail the aerosol motion can be different from the air motion and the statement is not strictly correct."

Response: The spectral characteristics of aerosol number concentration fluctuations approximate the spectral characteristics of molecular density fluctuations. Please see Line 175-176.

10) "L. 166-167, temperature is not a passive atmospheric constituent because buoyancy affects strongly the motion of air. Also "distribution" does not seem relevant but maybe just "small particles". Rather say that similarity of atmospheric aerosols and temperature can be assumed for the purpose."

Response: You are very kind and helpful for the manuscript. Please see Line 206-207.

11) "L. 173 "aerosol particles are continuously dispersed in the air", the meaning and purpose of this sentence is not clear."

Response: Although the aerosol particles are dispersed in the air, the macroscopic behavior of the gas-particle two-phase mixture is the same as if it is perfectly continuous in structure and physical quantities, such as the mass and refractive index associated with the matter contained within a given small volume, which will be regarded as being spread continuously over that volume. We modified the manuscript. Please see Line 213-216.

12) "L. 192, Correct R_{MN}"

Response: We Modified it.

13) "L. 209-212, please provide reference and/or explanation for the relation between the high/low frequency fluctuations and the real/imaginary parts of the AERI." Response: and can be measured by a specially made LAS (Yuan et al., 2015). After a spherical wave propagates over a distance in a turbulent atmosphere, the light intensity on the receiving end will fluctuate. When the attenuation caused by scattering and absorption along the propagation path is very weak, light intensity fluctuation depends on the variation of the real part of the AERI along the propagation path. When the attenuation caused by scattering and absorption along the propagation path is relatively strong, the light intensity fluctuation is also related to the fluctuation of the imaginary part of the AERI along the propagation path. With the spectral analysis method, the LAS light intensity fluctuations can be separated into the contributions of the real and imaginary parts of the AERI. The input of the real part of the AERI corresponds to the high frequencies, whereas the participation of the imaginary part of the AERI corresponds to the low frequencies, suggesting that the variances resulting from the real and imaginary parts are independent. Therefore, the light intensity variances induced by the real and imaginary parts can be detected separately at high frequencies and low frequencies from the LAS measurements (Yuan et al. 2015). We added the explanation. Please Line 257-271.

14) "L. 225, turbulent fluctuations of what?"

Response: Temperature

15) "L. 297, e.g. stands for "for example", not relevant here."

Response: Namely

16) "L. 309-310. The method for judging.. sentence difficult to follow, rephrase."

Response: The 6-point moving average is done for the trend.

17) "L. 315, how was "mean of the adjacent difference" defined, based on the moving

average or how? Improve wording of the sentence."

Response: The difference between adjacent moments is denoted and abbreviated as AMD. We have modified the sentences. We modified the paragraph. Please Line 373-385.

18) "L. 321-321, is the exact shape of the spectrum relevant? Or the method relies purely on the Kolmogorov's power laws of the spectra?"

Response: The theoretical expression for the relation between light scintillation and the structure parameter are based on von Karman spectrum. It is assumed that the actual turbulence is accord with von Karman spectrum, and then the parameters are calculated. The deviation from the assumption will cause the error. We modified the sentence. Please see Line 619-620.

19) "L. 328, "heavy pollution weather conditions "is a bit weird, please rephrase"

Response: under heavy pollution weather conditions modified as during heavy pollution episodes (HPEs). Please see Line 397.

20) "L. 361, rather the wind direction varied throughout day? there was no dominant wind direction"

Response: The wind direction has diurnal variation characteristics, which are related to the sea-land breeze, valley wind and urban heat island circulation which may exist under the control of weak weather system. Please see Line 433-435.

21) "L. 368-374. The "free convection" conditions are not always easily satisfied. Free convection means that the buoyancy-driven turbulence dominates over mechanical turbulence and this is not just the unstable conditions but the free convective limit of the unstable conditions. Please clarify and evaluate the uncertainties introduced by such assumption."

Response: At the CAMS site, local turbulence and local stability parameter measurements cannot be implemented. So we can only choose an alternative, and used the meteorological data (temperature) measured at nearby observation points, then based on the free convection assumption (using Equ. 12) the aerosol fluxes at the CAMS site were calculated. We (Yuan et al, 2015) conducted a test experiment for vertical aerosol flux in Hefei City, China, using free convection assumptions and local similarity theories to calculate aerosol fluxes. Comparison of the calculation results of the two methods shows that precarious condition, -0.15 < (z-zd) / L < 0, accounts for about 62 % of the time, and the relative difference is about 5%. Under weak unstable and stable condition, the relative error is about 15%. Although the relative error is a little significant under weak unstable stable stratification conditions, the absolute difference in flux is still small. We added some explanation. Please see L445-451.

22) "L. 421, southerly wind conditions"

Response: southerly wind

23) "Figures 3-6, the square value of the structure parameter is plotted according to label in y-axis of the relevant subplots."

Response: Regarding the naming of Cn and Cn2, I checked it. In the book of Monin and Yaglom (1975) and the book of Tatarskii (1961), there is no name for Cn and Cn2. In later books, different books have different names. For example, in Ishimaru (1978), Cn is called a structure parameter, and Cn2 is also called a structure parameter. In Andrews and Phillip (2005), Cn2 is called a structural parameter. Considering that we have always called Cn2 as a structural parameter (Yuan et al.,2015; Yuan et al. 2016), and in the application of Cn2 it generally appears in the form of Cn2, I will refer to Cn2 as a structural parameter in this manuscript. Andrews, L. C., and Phillips, R. L.: Laser beam propagation through random media, SPIE, 2005. Monin, A. S., and Yaglom, A. M.: Statistical fluid mechanics: Mechanics of turbulence, MIT Press, Cambridge, Massachusetts, 874 pp., 1975. Ishimaru, A.: Wave propagation and scattering in random media, Oxford University Press, Walton Street, Oxford OX2 6DP, 590 pp., 1978.

Tatarskii, V. I.: Wave Propagation in a Turbulent Medium, McGraw-Hill Book Company Inc., New York, 285 pp., 1961. Yuan, R., Luo, T., Sun, J., Liu, H., Fu, Y., and Wang, Z.: A new method for estimating aerosol mass flux in the urban surface layer using LAS technology, Atmospheric Measurement Techniques, 9, 1925-1937, 10.5194/amt-9-1925-2016, 2016. Yuan, R., Luo, T., Sun, J., Zeng, Z., Ge, C., and Fu, Y.: A new method for measuring the imaginary part of the atmospheric refractive index structure parameter in the urban surface layer, Atmospheric Chemistry and Physics, 15, 2521-2531, 10.5194/acp-15-2521-2015, 2015.

24) "Discussion and conclusions: how do the measured aerosol mass-fluxes compare with relevant literature values and/or earlier measurements and typical emission estimates? Please discuss this how to results contribute to understanding of pollution emissions."

Response: Compared to the results (Yuan et al. 2016) from Hefei, China, a small and medium-sized provincial capital city in East China, the measured aerosol mass-fluxes in Beijing are almost at the same amount. A series of measures and actions have been made for emission reduction in Beijing, and the primary emission is from vehicles. The difference in aerosol mass flux may be small. We added some discussion. Please Line 604-607.

Finally, the authors thank you for your constructive comments that help us to improve the clarity and the quality of the manuscript greatly. All the comments are answered and the modifications introduced in the revised manuscript correspondingly. We sincerely hope our answers can relieve doubts and give a better description of our work.

---

## Author Comment (AC2) · 29 Jun 2019

Authors reply to reviewer's comments:

Dear Anonymous Referee,

Thanks for your careful review of the manuscript. We read the reviewer's comments carefully, and have responded and taken all of the reviewer's comments into consideration and revised the manuscript accordingly. My detailed responses are as follows: Comments from Anonymous Referee #1:

[Figure]

Creative Commons BY license logo

East Asia, especially China, is facing heavy haze pollution in wintertime. Though many measurements on air pollutants have been extensively conducted across China, there is still a lack of flux data on pollutants, which may play a substantial role in haze formation. This study combines measurements on meteorological conditions, flux, as well as PM2.5 concentration in BTH region to derive aerosol vertical mass flux, and provide some observational insight on aerosol vertical flux under stable condition. Therefore, this manuscript adds to our current knowledge of aerosol vertical exchange and its im-pact on meteorology. However,I have some concerns about the methods/data analysis used in the study and the interpretation of results , and more in-depth analysis and discussion ought to be provided. I think this manuscript can be considered for publication only if the authors could adequately address the comments below.

"Major comment: 1) There are two observation sites, a rural site (GC site) and an urban site (CAMS site). The Monin-Obukhov similarity theory (MOST) is applied in rural site because the surface is homogenous. But in the urban site, the observation was within the urban roughness sublayer (3-5 mean building height), MOST is invalid due to the lack of constant-flux conditions, the local similarity theory should be used. In other words, the function or the parameters in the similarity relationship should be different for the rural and urban site."

Response: Indeed, in the urban site (CAMS site) the observation was within the ur-ban roughness sublayer, and the local similarity theory should be used to calculate the aerosol mass flux. But, if the local similarity theory is applied for calculation in our experiment, the local turbulence parameters and local stability parameters are required. At the CAMS site, these parameter measurements cannot be implemented due to actual conditions. So we can only choose an alternative, and used the meteorological data (temperature) measured at nearby observation points, then based on the free convection assumption (using Equation 12) the aerosol fluxes at the CAMS site were calculated. We (Yuan et al, 2015) conducted a test experiment for vertical aerosol flux in Hefei City, China, using free convection assumptions and local similarity theories to

calculate aerosol fluxes, respectively. Comparison of the calculation results of the two methods shows that very unstable condition, $-0.15 < (z-zd) / L < 0$, accounts for about 62 % of the time, and the relative difference is about 5%. Under weak unstable and stable condition, the relative error is about 15%. Although the relative error is a little large under weak unstable stable stratification conditions, the absolute difference in flux is still small. There is a weather tower in the north of Beijing. The weather tower is 6.1 km far from the CAMS site. The meteorological observation data from the weather tower show that the Monin-Oubhov similarity theory has a little significant error under stable condition, while the Monin-Oubhov similarity theory is still basically applicable in the case of unstable stratification (Liu et al. 2009). In the roughness sub-layers of other cities, under the condition of unstable stratification, the local similarity theory is similar to MOST (Zou et al. 2018, 2019). Urban meteorological observations show that the urban surface layer is almost always in an unstable stratification. Even if the city's upper levels are stable, it is nearly always unstable near the ground in the city (Li et al., 2007). All of this shows that our current treatment is reasonable. Please see L442-L463

Reference Li, X., Hu, F., and Shu, W.: Study on the characteristics of winter island heat islands in Beijing and the influence factors of strong and weak heat islands, Journal of the Graduate School of the Chinese Academy of Sciences, 4, 431-438, 2007. Liu Ximing, Hu Fei, Quan Lihong, Cao Xiaoyan, and Dou Junxia, 2009, Validation of the local similarity in urban boundary layer, Climatic and Environmental Research (in Chinese), 14(2): 183-191. Zou, J., Liu, G., Sun, J., Zhang, H., and Yuan, R.: The momentum flux-gradient relations derived from field measurements in the urban roughness sub-layer in three cities in China, Journal Of Geophysical Research-Atmospheres, 120, 10.1002/2015jd023909, 2015. Yuan, R., Luo, T., Sun, J., Liu, H., Fu, Y., and Wang, Z.: A new method for estimating aerosol mass flux in the urban surface layer using LAS technology, Atmospheric Measurement Techniques, 9, 1925-1937, 10.5194/amt-9-1925-2016, 2016. Roth, M.: Review of atmospheric turbulence over cities, Quarterly Journal of the Royal Meteorological Society, 126, 941-990, 10.1256/smsqj.56408,

2000.

2) "The function and parameters of the similarity relationship are not universal, the authors should explain why they use these function and parameters in the paper. For example, in Eq. 4, the authors said that they take the parameters b1 and b2 follow DeBruin et al., 1995. But in DeBruin et al., 1995, it said that "For stable conditions there is no consensus on the universal function", b1=5,b2=0 were found by DeBruin et al., 1993, and "the scatter was very large". So DeBruin may not be the best choice. Especially, in Yuan et al., 2016, the parameter b1 and b2 follows Wyngaard et al., 1971., which is very different from DeBruin et al., 1993. When b1 and b2 follow DeBruin et al., 1995, it means that $\eta(\xi)$ stays constant with stability; but when b1 and b2 follow Wyngaard et al., 1971, it means that $\eta(\xi)$ changes constant with stability. The author should explain why they choose DeBruin et al., 1995."

Response: In addition to DeBruin et al., 1993 and 1995, there are a number of schemes that are used to parameterize the near-surface temperature structure parameter CT2. Available data, such as CT2, u* and T*, were used to calculate the difference between schemes and actual data. The scheme with the smallest difference was selected. The experiment of Yuan et al (2016) was conducted over the urban surface. The scheme of DeBruin et al (1995) was used for processing of these data for unstable conditions, and the scheme of Wyngaard et al.(1971) was used for stable condition. When the free convection approximation is satisfied, the approximate expression given by Wyngaard et al., 1971 was used. The current GC site is a rural site with a flat underlying surface, where similarity theory can be applied. The parameters T* and u* were obtained from temperature-wind profiles from a tower in GC site. After comparing several parameterization schemes, we found that taking b1=5 and b2=0 was the best match with the actual results. So we used this scheme. Details are given below. The parameterizing scheme for the near-surface temperature structural parameter CT2 can be expressed by the formula in the literature (Wyngaard et al., 1971), i.e.

Equation 1 (1)

where z is the measurement height, d is the zero-displacement height, $\xi=(z-d)/L$ is the nondimensional stability parameter, L is the Monin-Obukhov (M-O) length and defined as. Usually, can be,

Equation 2 $0 \geq (z-d)/L$ (2)

Equation 3 $0 \leq (z-d)/L \leq 2$ (3)

(See Fig.2 below for the complete formula. Eqs.(1)-(3) )

Five coefficients a1, a2, b1, b2, e1 in Eqs. (2) (3) were decided by different researchers, shown in Table 1.

Table1 Five coefficients in universal function (See Fig.4 below for Table1.)

Schemes 1, 2 and 4 were widely used, so the three were used to calculate flux for comparison in our experiment. Sensible heat flux can be calculated as,

Equation 4 i=1,2,4, for scheme number. (4)

and compared with

Equation 5 (5)

(See Fig.3 below for the complete formula. Eqs.(4)-(5) )

The variables and can be obtained from 3-D sonic anemometer or temperature-wind profiles. Comparison of sensible heat flux of Eqs. (4) and (5) is equivalent to the comparison between and in Eq. (1) Aerosol flux measurement was conducted in Hefei, China (Yuan et al. 2016), and CT2 was deduced from a LAS and L were deduced from an EC system. Taking a1=4.9, a2=9, b1=4.9, b2=2.75, and e1=0 gives the minimal difference between Eq. (4) and Eq. (5). For the experiment at the GC site, CT2 was deduced from a LAS and L were deduced from wind profile and temperature profile. Comparisons of sensible heat flux between Hsi and Hs0 were done and shown in Fig. 1.

Figure 1 Comparisons between Hsi and Hs0

(a)(b)(c) corresponding to scheme 1,2,4 respectively and statistical results are given on the lower right. (See Fig.1 below for Figure 1 )

From comparisons in Fig. 1, Scheme 4 was selected to calculate flux in the current manuscript. The effect of the footprint is not considered in our experiment. Please see Line 194-Line 200. Reference Andreas, E. L.: Estimating cn2 over snow and sea ice from meteorological data, Journal of the Optical Society of America a-Optics Image Science and Vision, 5, 481-495, 10.1364/josaa.5.000481, 1988. Wyngaard, J. C., Izumi, Y., and Collins, S. A.: Behavior of refractive-index-structure parameter near ground, J. Opt. Soc. Am., 61, 1646-1650, 10.1364/josa.61.001646, 1971. De-Bruin, H. A. R., vandenHurk, B., and Kohsiek, W.: The scintillation method tested over a dry vineyard area, Boundary-Layer Meteorology, 76, 25-40, 1995. Debruin, H. A. R., Kohsiek, W., and Vandenhurk, B.: A verification of some methods to determine the fluxes of momentum, sensible heat, and water-vapor using standard-deviation and structure parameter of scalar meteorological quantities, Boundary-Layer Meteorology, 63, 231-257, 1993. Andreas, E. L.: 1989, 'Two-Wavelength Method of Measuring Path-Averaged Turbulent Surface Heat Fluxes', J. Atmos. Oceanic Tech. 6, 280–292. Maronga, B.: Monin-Obukhov Similarity Functions for the Structure Parameters of Temperature and Humidity in the Unstable Surface Layer: Results from High-Resolution Large-Eddy Simulations, Journal of the Atmospheric Sciences, 71, 716-733, 10.1175/jas-d-13-0135.1, 2014. Li, D., Bou-Zeid, E., and De Bruin, H. A. R.: Monin-Obukhov Similarity Functions for the Structure Parameters of Temperature and Humidity, Boundary-Layer Meteorology, 145, 45-67, 10.1007/s10546-011-9660-y, 2012. Hartogensis, O. K., and H. A. R. De Bruin, 2005: Monin–Obukhov similarity functions of the structure parameter of temperature and turbulent kinetic energy dissipation rate in the stable boundary layer. Bound.-Layer Meteor., 116, 253–276. Zhang, H., and Zhang, H.: Comparison of Turbulent Sensible Heat Flux Determined by Large-Aperture Scintillometer and Eddy Covariance over Urban and Suburban Areas, Boundary-Layer Meteorology, 154, 119-136, 10.1007/s10546-014-9965-8, 2015. Braam, M., Beyrich, F., Bange, J., Platis, A., Martin, S., Maronga, B., and Moene, A. F.: On the Discrepancy in Simultaneous Observations of the Structure Parameter of Temperature Using Scintillometers and Unmanned Aircraft, Boundary-Layer Meteorology, 158, 257-283, 10.1007/s10546-015-0086-9, 2016. Lee, S.-H., Lee, J.-H., and Kim, B.-Y.: Estimation of Turbulent Sensible Heat and Momentum Fluxes over a Heterogeneous Urban Area Using a Large Aperture Scintillometer, Advances In Atmospheric Sciences, 32, 1092-1105, 10.1007/s00376-015-4236-2, 2015. Thiermann, V., and Grassl, H.: The measurement of turbulent surface-layer fluxes by use of bichromatic scintillation, Boundary-Layer Meteorology, 58, 367-389, 10.1007/bf00120238, 1992. Li, X., Gao, Z., Li, Y., and Tong, B.: Comparison of Sensible Heat Fluxes Measured by a Large Aperture Scintillometer and Eddy Covariance System over a Heterogeneous Farmland in East China, Atmosphere, 8, 10.3390/atmos8060101, 2017. Yuan, R., Luo, T., Sun, J., Liu, H., Fu, Y., and Wang, Z.: A new method for estimating aerosol mass flux in the urban surface layer using LAS technology, Atmospheric Measurement Techniques, 9, 1925-1937, 10.5194/amt-9-1925-2016, 2016.

3) "In L359 The conventional meteorological parameters were measured at 20m above the ground surface. But in L275, the author said that the measurement heights of temperature and wind speed were 1.5 m and 10 m at CAMS site (Beijing). It should be clear which data were used to calculate the aerosol flux. Because the average height of the building was 24m in CAMS site, and LAS was located at 43meters. The temperature measured at 1.5m within the canopy layer is different from 43m above the canopy layer, and the calculation of aerosol fluxes from Eq. 12 was badly influenced."

Response: There are a few errors in depicting measurement height. The conventional meteorological parameters are measured on the same roof, 20 meters away from the receiving end and in the northwest direction of the receiving end. The measurement heights were 1.5 m and 10 m above the roof for air temperature and wind speed. Please see L336-L339.

4) "Another issue that the authors need to address is the assumption between AERI( atmospheric equivalent refractive index) and aerosol mass concentration as well as aerosol adsorption. First, there do exist some light-absorbing trace gases in the atmosphere, which may influence AERI significantly. Second, aerosol absorption generally contributes a relatively small part of the extinction. By contrast, scattering components like sulfate and organic matters dominate aerosol extinction during haze pollution episode, especially under high humidity. Last but not at least, aerosol extinction is also closely related to the number concentration and size distribution, which need to be considered here. I do not think it is technically robust to simply get the relationship between the imaginary part of the AERI and the atmospheric aerosol mass concentration in Eq.6."

Response: The light wavelength is 0.620 $\mu$m. This wavelength is only weakly absorbed by O3; therefore, the observed absorption is primarily due to aerosol (Brion et al., 1998; Lou et al., 2014; Nebuloni, 2005). Aerosol extinction is also closely related to the number concentration, size distribution, and refractive index of aerosol particles, so there is not a simple linear relationship between the imaginary part of the AERI and the atmospheric aerosol mass concentration in Eq.6. The variations in the ratio of the aerosol mass concentration to the imaginary part of the AERI will introduce errors into the aerosol mass flux measurements. RMN should be obtained by simultaneously measuring Ma and the imaginary part of the AERI at the same location with the LAS so that real-time RMN can be obtained. For GC site and CAMS site, measuring positions of PM10 and visibility are a little far from LAS measurement. So a constant ratio RMN is more representative than a simultaneous value. Our experiment conducted in Hefei (Yuan et al. 2016) showed that the linear correlation coefficient between PM10 and the imaginary part of the AERI is 0.94. This indicates that constant is a reasonable assumption for a given location with a dominant aerosol type, such as urban aerosols. Of course, when measurements for aerosol flux using a LAS, PM10 and visibility are performed together, simultaneous value for RMN is better. Please see L234-L236 and L408-L422. Reference: Brion, J., Chakir, A., Charbonnier, J., Daumont, D., Parisse,

C., and Malicet, J.: Absorption spectra measurements for the ozone molecule in the 350-830 nm region, J. Atmos. Chem., 30, 291-299, 10.1023/a:1006036924364, 1998. Lou, S., Liao, H., and Zhu, B.: Impacts of aerosols on surface-layer ozone concentrations in China through heterogeneous reactions and changes in photolysis rates, Atmos. Environ., 85, 123-138, 10.1016/j.atmosenv.2013.12.004, 2014. Nebuloni, R.: Empirical relationships between extinction coefficient and visibility in fog, Appl. Opt., 44, 3795-3804, 10.1364/ao.44.003795, 2005. Yuan, R., Luo, T., Sun, J., Liu, H., Fu, Y., and Wang, Z.: A new method for estimating aerosol mass flux in the urban surface layer using LAS technology, Atmospheric Measurement Techniques, 9, 1925-1937, 10.5194/amt-9-1925-2016, 2016.

"Minor issues: Some statements in this manuscript are very hard to follow. Language editing is needed for improving the accuracy of language as well as overall readability."

Response: We've tried our best to improve the English writing in the revised manuscript, but also a native English speaker reviewed the revised version of the manuscript.

1) "Line 43: Please rephrase 'heavy pollution weather'"

Response: heavy pollution environment Please see Line 45-46.

2) "Line 48: 'few studies' should be 'few study'"

Response: We modified it.

3) "Line 72: what is the boundary layer box model? Usually box model is zero-dimensional."

Response: The boundary layer box model means the boundary layer is taken as a box. The box is filled within duration $\tau$ with flux F, and then flux F can be estimated by the boundary layer box model: F=C*HBL/$\tau$ where C was the concentration measured at 30 m height, HBL was the measured boundary layer height averaged over the sample duration, and $\tau$ was boundary layer filling time. More details is in Ceburnis et al. (2016).

I add an explanation. Please see Line 77-78. Referenceïij$\check{Z}$ Ceburnis, D., Rinaldi, M., Ovadnevaite, J., Martucci, G., Giulianelli, L., and O'Dowd, C. D.: Marine submicron aerosol gradients, sources and sinks, Atmospheric Chemistry and Physics, 16, 12425-12439, 10.5194/acp-16-12425-2016, 2016.

4) "Line 106: should be 'makes it possible'"

Response: We modified it in Line 113.

5) "Eq. 11: replace z with (z-d)"

Response: We modified it. Please see Line 276.

6) "Line 304: More detail needed, not "personal experience.""

Response: We specified personal experience as " trend comparison for same variables between different heights and different locations." Please see L367-368.

7) "Line 378: weakly unstable is not free convection. The free convection assumption was not satisfied at night."

Response: When the free convection assumption is applied to weakly unstable condition at night, the assumption will result in an some uncertaintie. In many cases, similar to the LAS-derived sensible heat flux, we can only choose free convection assumption to obtain flux. Under stable conditions or weakly unstable condition, the value of the flux data is small and does not cause significant error. Of course, a better approach is to get u* and T* from meteorological variable and calculate aerosol flux according to Eq. (10). Please see L445-451.

Finally, the authors thank you for your constructive comments that help us to improve the clarity and the quality of the manuscript greatly. All the comments are answered and the modifications introduced in the revised manuscript correspondingly. We sincerely hope our answers can relieve doubts and give a better description of our work.

Please also note the supplement to this comment:
https://www.atmos-chem-phys-discuss.net/acp-2018-1265/acp-2018-1265-AC2-supplement.pdf

———————————————————

statistical results

| No | Slope | $R^2$ | RMSE($Wm^{-2}$) |
|---|---|---|---|
| 1 | 0.91 | 0.86 | 14.6 |
| 2 | 0.87 | 0.86 | 14.1 |
| 4 | 0.98 | 0.87 | 14.3 |

**Fig. 1.** Figure 1

$$\frac{C_T^2(z-d)^{2/3}}{T_*^2}=\eta(\frac{z-d}{L}) \tag{1}$$

where z is the measurement height, d is the zero-displacement height, ξ=(z-d)/L is the nondimensional stability parameter, L is the Monin-Obukhov (M-O) length and defined as $L=\frac{\bar{T}u_*^2}{\kappa g T_*}$. Usually, $\eta(\frac{z-d}{L})$ can be,

$$\eta(\frac{z-d}{L})=a_1[1-a_2\frac{z-d}{L}]^{-2/3} \quad 0\geq(z\text{-}d)/L \tag{2}$$

$$\eta(\frac{z-d}{L})=b_1[1+b_2(\frac{z-d}{L})^{e_1}] \quad 0\leq(z\text{-}d)/L\leq2 \tag{3}$$

**Fig. 2.** Equations 1-3

Schemes 1,2 and 4 were widely used, so the three were used to calculate flux for comparison in our experiment. Sensible heat flux can be calculated as,

$$H_S{}^i = C_p \rho u_* \sqrt{\frac{C_T^2 z^{2/3}}{\eta(z/L)}} \quad \text{i=1,2,4,for scheme number.} \quad (4)$$

and compared with

$$H_S{}^0 = C_p \rho u_* T_* \quad (5)$$

The variables $u_*$ and $T_*$ can be obtained from 3-D sonic anemometer or temperature-wind profiles. Comparison of sensible heat flux of Eqs. (4) and (5) is equivalent to the comparison between $\sqrt{\dfrac{C_T^2 z^{2/3}}{\eta(z/L)}}$ and $T_*$ in Eq. (1)

**Fig. 3.** Equations 4-5

Table1 Five coefficients in universal function

| Scheme no | $a_1$ | $a_2$ | $b_1$ | $b_2$ | $e_1$ | References |
|---|---|---|---|---|---|---|
| **1** | **4.9** | **7** | **4.9** | **2.75** | **1** | Wyngaard,1971; He_2018 |
| **2** | **4.9** | **6.1** | **4.9** | **2.2** | **2/3** | Andreas(1988,1989),Zhang(2015), Braam_2016,Lee_2015, Li,2017 |
| 3 | 4.9 | 7 | 6.34 | 7 | 1 | Thierrnann 等 1992 |
| **4** | **4.9** | **9** | **5** | **0** | **1** | De Bruin 等,1993,1995 |
| 5 | - | - | 4.9 | 2.4 | 2/3 | Hartogensis 等,2005 |
| 6 | 6.1 | 7.6 | | | | Maronga_2014 |
| 7 | 6.7 | 14.9 | 4.5 | 1.3 | 2/3 | Li et al.,2012 |
| 8 | | | 4.7 | 1.6 | 2/3 | Hartogensis,2005 |

**Fig. 4.** Table1

---

## Referee Report (RR1)

General comments

The authors have carefully considered all comments raised by referees and the manuscript has much improved. The only thing that I am not fully confident that the free convection assumption is applicable over majority of observations at CAMS site (there is probably a typo in the response letter where the range of conditions $-0.15 < (z-z_d)/L < 0$ is called very unstable; this corresponds to conditions from near-neutral to moderately unstable). However, the authors have quantified the uncertainty related to this assumption and I am ok with the response.

Overall I am positive for publication. There are still number of typos and unclear wordings in the manuscript. Below is a list of some (not certainly exhaustive). The manuscript would still require careful reading and correction, and removing some repetitions.

1. Line 67, replace "much high" with just "high"
2. L 77-78, do not repeat "and the boundary layer is taken as a box".
3. L. 108, better use "are not representative of wider area", L. 109, would it make sense "larger spatial representation" instead of accurate? Measurement accuracy is a different concept but here we talk mostly about spatial representativeness.
4. L. 112 "atmospheric surface layer similarity theory" (or Monin-Obukhov…)
5. L. 114 replace repetition "light propagation theory and…" with "the same principles"
6. L. 136: "PM2.5 dominated by wind" is loose wording. Perhaps "PM2.5 high concentration levels are caused…"
7. L. 138: What rising process? Probably you mean elevation of PM2.5 concentrations. The same in L. 144 (for the rising…)
8. L. 149 what are "vertical aerosols"? improve
9. L. 152 how near-ground cooling effect is caused by atmospheric circulation and vertical mixing? Perhaps you mean lack of vertical mixing?
10. L. 155 instead of "mostly" better "predominantly"?
11. L. 167, Instead of "argument" use "principles"
12. L. 198, it is not clear from text if the choice of coefficients was based on the experimental data of this study or some other. Also, better "based on minimal difference"?
13. L. 244, "relatively small variations in particle size": particle distributions are usually over very wide range of sizes; do you mean here small variations in size distributions?
14. Line 388, instead of "supplementation" use "gap-filling"
15. L. 389, verb is missing in the first part of sentence, add "exist"?
16. L. 390, more clear "impact of the deviation of the shape of spectrum from…"?
17. L. 406, this is not truly scatter diagram because bin-averaging has been performed. Call it relationship plot or similar.
18. L. 408: use "The fitted line"
19. L. 420: $R_{MN}$
20. L. 432: Better "Moderately strong" because the wind speed values were still fairly moderate
21. L. 433: better "has diurnal variation, which is related"
22. L. 488: I would skip the badly worded sentence. And in general 15% relative error is fairly small considering large uncertainties of aerosol fluxes in general.

23.    L. 453: This is badly worded sentence. The Monin-Obukhov has a significant error… in terms of what? It is not the theory that has error but its applicability under these conditions. Revise this sentence.
24.    I would suggest to move the whole paragraph (L. 451-462) into methods section (e.g. after line 281)
25.    L. 483, explain what was the difference or remove "except that there is a slight difference"
26.    L. 589: "from the ground": maybe better "from near-ground emission sources" because pollution source below the observation level (including buildings) contribute to emissions.

---

## Author Response (AR2)

Authors reply to reviewer's comments:

Dear Anonymous Referees,

Thanks for your careful review of the manuscript. We read the reviewer's comments carefully, and have responded and taken all of the reviewer's comments into consideration and revised the manuscript accordingly. My detailed responses are as follows:

**The authors have carefully considered all comments raised by referees and the manuscript has much improved. The only thing that I am not fully confident that the free convection assumption is applicable over majority of observations at CAMS site (there is probably a typo in the response letter where the range of conditions $-0.15 < (z-z_d)/L < 0$ is called very unstable; this corresponds to conditions from near-neutral to moderately unstable). However, the authors have quantified the uncertainty related to this assumption and I am ok with the response.**

Response:
We made a typo in the response letter. The range of conditions $(z-zd)/L < -0.15$ is called very unstable, which is in accordance with our previous paper (Yuan et al. 2016).

1. **Line 67, replace "much high" with just "high"**
Response: We modified the manuscript as suggestion.
2. **L 77-78, do not repeat "and the boundary layer is taken as a box".**
Response: We modified the manuscript as suggestion and deleted the words.
3. **L. 108, better use "are not representative of wider area", L. 109, would it make sense "larger spatial representation" instead of accurate? Measurement accuracy is a different concept but here we talk mostly about spatial representativeness.**
Response: We modified the manuscript as suggestion.
4. **L. 112 "atmospheric surface layer similarity theory" (or Monin-Obukhov…)**
Response: We modified the manuscript as suggestion
5. **L. 114 replace repetition "light propagation theory and…" with "the same principles"**
Response: We modified the manuscript as suggestion
6. **L. 136: "PM2.5 dominated by wind" is loose wording. Perhaps "PM2.5 high concentration levels are caused…"**
Response: We modified the manuscript as suggestion
7. **L. 138: What rising process? Probably you mean elevation of PM2.5 concentrations. The same in L. 144 (for the rising…)**
Response: We mean elevation of PM2.5 concentrations and modified the manuscript as suggestion

8.  **L. 149 what are "vertical aerosols"? improve**

Response: "vertical aerosols" means "aerosol particles by vertical transport". We modify the sentence.

9.  **L. 152 how near-ground cooling effect is caused by atmospheric circulation and vertical mixing? Perhaps you mean lack of vertical mixing?**

Response: Yes, a near-ground cooling effect results in the lack of vertical mixing in the near-surface layer. We modified the manuscript as suggestion.

10. **L. 155 instead of "mostly" better "predominantly"?**

Response: We modified the manuscript as suggestion.

11. **L. 167, Instead of "argument" use "principles"**

Response: We modified the manuscript as suggestion

12. **L. 198, it is not clear from text if the choice of coefficients was based on the experimental data of this study or some other. Also, better "based on minimal difference"?**

Response: We modified the manuscript as suggestion. The choice of coefficients was based on the experimental data of this study.

13. **L. 244, "relatively small variations in particle size": particle distributions are usually over very wide range of sizes; do you mean here small variations in size distributions?**

Response: Yes, we mean here small variations in size distributions

14. **Line 388, instead of "supplementation" use "gap-filling"**

Response: We modified the manuscript as suggestion.

15. **L. 389, verb is missing in the first part of sentence, add "exist"?**

Response: We modified the manuscript as suggestion.

16. **L. 390, more clear "impact of the deviation of the shape of spectrum from…"?**

Response: We modified the manuscript as suggestion.

17. **L. 406, this is not truly scatter diagram because bin-averaging has been performed. Call it relationship plot or similar.**

Response: We modified the manuscript as suggestion.

18. **L. 408: use "The fitted line"**

Response: We modified the manuscript as suggestion.

19. **L. 420: $R_{MN}$**

Response: We modified the manuscript as suggestion.

20. **L. 432: Better "Moderately strong" because the wind speed values were still fairly moderate**

Response: We modified the manuscript as suggestion.

21. **L. 433: better "has diurnal variation, which is related"**

Response: We modified the manuscript as suggestion.

22. **L. 488: I would skip the badly worded sentence. And in general 15% relative error is fairly small considering large uncertainties of aerosol fluxes in general.**

Response: We modified the manuscript as suggestion and deleted the sentence..

23. **L. 453: This is badly worded sentence. The Monin-Obukhov has a significant error… in terms of what? It is not the theory that has error but its applicability under these conditions. Revise this sentence.**

Response: L. 453 in original manuscript is modified as, "applicability of the Monin-Oubhov similarity theory under stable condition causes a significant error for $T*$ or $u*$,"

24. **I would suggest to move the whole paragraph (L. 451-462) into methods section (e.g. after line 281)**

Response: We modified the manuscript as suggestion.

25. **L. 483, explain what was the difference or remove "except that there is a slight difference"**

Response: We modified the manuscript as suggestion and remove the sentence.

26. **L. 589: "from the ground": maybe better "from near-ground emission sources" because pollution source below the observation level (including buildings) contribute to emissions.**

Response: We modified the manuscript as suggestion.

We also modified some typo errors. Please see the marked document.

Finally, the authors thank the two referees for their constructive comments that help us to improve the manuscript greatly.